# Quantification of anti-parasite and anti-disease immunity to malaria as a function of age and exposure

Isabel Rodriguez-Barraquer[1]*, Emmanuel Arinaitwe[2], Prasanna Jagannathan[3], Moses R Kamya[4], Phillip J Rosenthal[1], John Rek[2], Grant Dorsey[1], Joaniter Nankabirwa[2], Sarah G Staedke[5], Maxwell Kilama[2], Chris Drakeley[5], Isaac Ssewanyana[2], David L Smith[6], Bryan Greenhouse[1]

[1]Department of Medicine, San Francisco General Hospital, University of California, San Francisco, San Francisco, United States; [2]Infectious Diseases Research Collaboration, Kampala, Uganda; [3]Department of Medicine, Stanford University, Stanford, United States; [4]Department of Medicine, Makerere University College of Health Sciences, Kampala, Uganda; [5]London School of Hygiene and Tropical Medicine, London, United Kingdom; [6]Institute of Health Metrics and Evaluation, University of Washington, Seattle, United States

**Abstract** Fundamental gaps remain in our understanding of how immunity to malaria develops. We used detailed clinical and entomological data from parallel cohort studies conducted across the malaria transmission spectrum in Uganda to quantify the development of immunity against symptomatic *P. falciparum* as a function of age and transmission intensity. We focus on: anti-parasite immunity (i.e. ability to control parasite densities) and anti-disease immunity (i.e. ability to tolerate higher parasite densities without fever). Our findings suggest a strong effect of age on both types of immunity, not explained by cumulative-exposure. They also show an independent effect of exposure, where children living in moderate/high transmission settings develop immunity faster as transmission increases. Surprisingly, children in the lowest transmission setting appear to develop immunity more efficiently than those living in moderate transmission settings. Anti-parasite and anti-disease immunity develop in parallel, reducing the probability of experiencing symptomatic malaria upon each subsequent *P. falciparum* infection.
DOI: https://doi.org/10.7554/eLife.35832.001

*For correspondence: isabel.rodriguez-barraquer@ucsf.edu

**Competing interests:** The authors declare that no competing interests exist.

## Introduction

The last decades have seen substantial declines in malaria transmission in sub-Saharan Africa that are largely attributable to increased access to effective control measures, including insecticide-treated bednets, indoor residual spraying of insecticide and artemisinin-based combination therapy (*Bhatt et al., 2015*; *World Health Organization, 2016*). In settings where transmission has been low, increased access to effective control interventions opens the possibility for malaria elimination. In highly endemic settings, however, there are concerns around the potential impact of failing to sustain interventions that reduce but do not stop transmission. Short-term decreases in malaria incidence due to reductions in transmission could be offset over time by reductions in population immunity to malaria resulting from lower exposure to parasites (*Filipe et al., 2007*; *Smith et al., 2001*; *Snow et al., 1997*).

Gradual acquisition of immunity against symptomatic malaria (also referred to as clinical immunity) is a key driver of the epidemiology of malaria in endemic settings, where the incidence of disease typically peaks in early childhood and then declines, while the prevalence of detectable

**eLife digest** Malaria kills around 500,000 children every year. The disease occurs when an infected mosquito bites a human and passes on a *Plasmodium* parasite. One parasite in particular, *Plasmodium falciparum*, is responsible for most malaria-related deaths across the globe. A person can be infected by *P. falciparum* many times throughout their life. However, after children have had multiple infections, they become less likely to develop symptoms of malaria, such as high fever. In other words, they gradually acquire immunity.

This immunity to malaria can come in two forms: "anti-parasite immunity", where the body fights the parasites and keeps their numbers low; and "anti-disease immunity", where the body is more likely to tolerate an infection without developing a fever. To date, scientists do not fully understand how either kind of immunity arises in children. Is it because they have simply been exposed to more malaria? Or does being older and having a more mature immune system also help?

Now, Rodriguez-Barraquer et al. have followed more than 1,000 children living in places with high, moderate and low rates of malaria infection in Uganda. Over three years, regular blood samples were taken to see if the children were infected with *P. falciparum*. Mosquitoes were also collected from their houses to estimate how often the children were being bitten and infected. Using this information, Rodriguez-Barraquer et al. studied the different factors that affect how children develop anti-parasite and anti-disease immunity.

Both types of immunity develop differently in places with high, moderate and low rates of infection, so being infected multiple times is important. Yet, the findings also show that growing older itself contributes to the development of immunity regardless of how often a child is infected.

Children who get infected most often – in other words, those living in houses with the most mosquitoes – develop immunity faster than those who get infected at a moderate rate. Unexpectedly, however, children living in places with low rates of infection also develop immunity faster than those living in places with moderate rates.

Understanding how children acquire immunity to malaria is important for people trying to control the disease. These results suggest that reducing rates of infection to very low levels may not interfere with development of immunity and may even improve it. However, future research should see if these findings apply to other parts of the world as well, and, if so, why children develop immunity faster in places with lower rates of malaria infection.
DOI: https://doi.org/10.7554/eLife.35832.002

asymptomatic parasitemia increases throughout childhood before declining in adulthood (*Griffin et al., 2015*; *Reyburn et al., 2005*; *Okiro et al., 2009*; *Carneiro et al., 2010*; *Idro et al., 2006*; *Roca-Feltrer et al., 2010*; *Rodriguez-Barraquer et al., 2016*). While these epidemiologic patterns have been described across the transmission spectrum, there are still many fundamental gaps in our understanding of the factors driving the development of immunity, and of the independent roles of age and repeated infection. One reason it has been challenging to study immunity to malaria is that there are currently no agreed upon reliable and quantifiable immune correlates of protection that can be measured in epidemiological studies (*Valletta and Recker, 2017*; *Fowkes et al., 2010*). In addition, there are few available datasets that include both detailed clinical data and independent metrics of exposure at the individual level.

Here, we use data from three parallel cohort studies conducted across the spectrum of malaria transmission in Uganda to model and quantify the development of immunity against symptomatic malaria as a function of transmission intensity and age. A key strength of these studies is that they involved detailed clinical and entomological surveillance of all study households. We focus on two specific types of immunity: anti-parasite immunity (i.e. the ability to control parasite densities upon infection) and anti-disease immunity (i.e. the ability to tolerate higher parasite infections without developing objective fever), as they have been described as independent components of clinical immunity (*Struik and Riley, 2004*).

## Results

The three cohorts enrolled a total of 1021 children aged 6 months to 10 years from 331 randomly chosen households across the three study sites. This analysis was limited to data from 773 children who experienced at least one patent *P. falciparum* infection between August 2011 and November 2014. *Table 1* summarizes the general characteristics of the participants included in this analysis.

Participants living in Nagongera experienced the highest incidences of symptomatic malaria (median 2.6 episodes per person year), followed by those living in Kihihi (median 1.6 episodes per person year) and Walukuba (median 0.6 episodes per person year) (*Table 1* and *Figure 1*). These incidences were consistent with results from monthly entomological surveys conducted in all cohort households, with significantly higher annual entomological inoculation rates (aEIR) recorded in Nagongera (median 51 infectious bites per year, range 10–582) as compared to Kihihi (median 8 infectious bites per year, range 4–47) and Walukuba (median 2 infectious bites per year, range 1–8). Interestingly, prevalence of asymptomatic parasitemia did not follow this same relationship; the prevalence of asymptomatic parasitemia was highest in Nagongera, and prevalences in the lower transmission sites were similar.

### aEIR as a metric of individual exposure

To assess whether entomological metrics were a good indicator of individual exposure to *P. falciparum*, we correlated the measured annual EIRs (aEIR) for each household (*Figure 2a*) with estimates of the average individual hazard of infection (*Figure 2b*). Individual hazards were estimated by fitting time-to-event models to the incidence data from each site. We found a significant correlation between these two independent metrics of exposure across sites ($R^2$ = 0.47, p<0.001). aEIR explained less of the variance between individuals within each site: Nagongera ($R^2$ = 0.03, p=0.004); Kihihi ($R^2$ = 0.12, p<0.001); Walukuba (0.01, p=0.05).

### Anti-parasite immunity

Parasite densities developed upon infection decreased with increasing age in all settings and for both symptomatic (passive detection) and asymptomatic (detected during routine visits) infections. Despite the large variability in parasite densities recorded within and between individuals, this trend is evident in the raw data (*Figure 3a*). A trend toward lower parasite densities was also observed among individuals living in settings with higher aEIRs (Nagongera), as compared to settings with lower aEIR (Kihihi and Walukuba).

**Table 1.** Characteristics of the study participants.

| Characteristic | Nagongera | Kihihi | Walukuba |
|---|---|---|---|
| Number of households | 106 | 100 | 76 |
| Number of children | 329 | 305 | 139 |
| Female, *n* (%) | 151(46) | 150 (49) | 66 (47) |
| Mean age at enrollment, years (*sd*) | 4.4 (2.7) | 4.6 (2.6) | 4.3 (2.6) |
| Mean follow up time, months (range) | 23.5 (0, 38.8) | 24.4 (0.8, 38.8) | 22.1 (2.3, 3.9) |
| *Symptomatic malaria* | | | |
| Symptomatic Malaria episodes, *n* | 2447 | 1555 | 207 |
| Median number of symptomatic malaria episodes/child, n (range) | 6 (0, 29) | 4 (0, 30) | 1 (0. 12) |
| Median incidence of symptomatic malaria episodes ppy (range) | 2.6 (0, 10) | 1.6 (0, 15.2) | 0.6 (0, 5.1) |
| *Asymptomatic parasitemia* | | | |
| Asymptomatic parasitemia episodes, n | 955 | 331 | 145 |
| Median number of asymptomatic parasitemia episode/child, n (range) | 2 (0, 12) | 0 (0, 11) | 1 (0, 10) |
| Median prevalence of asymptomatic parasitemia (range) | 0.12 (0.07–0.17) | 0.05 (0.02–0.10) | 0.07 (0.03–0.11) |
| *Household malaria exposure* | | | |
| Household aEIR, median (range) | 51 (10–582) | 7.7 (3.6–47) | 2.1 (1.5–8.1) |

DOI: https://doi.org/10.7554/eLife.35832.003

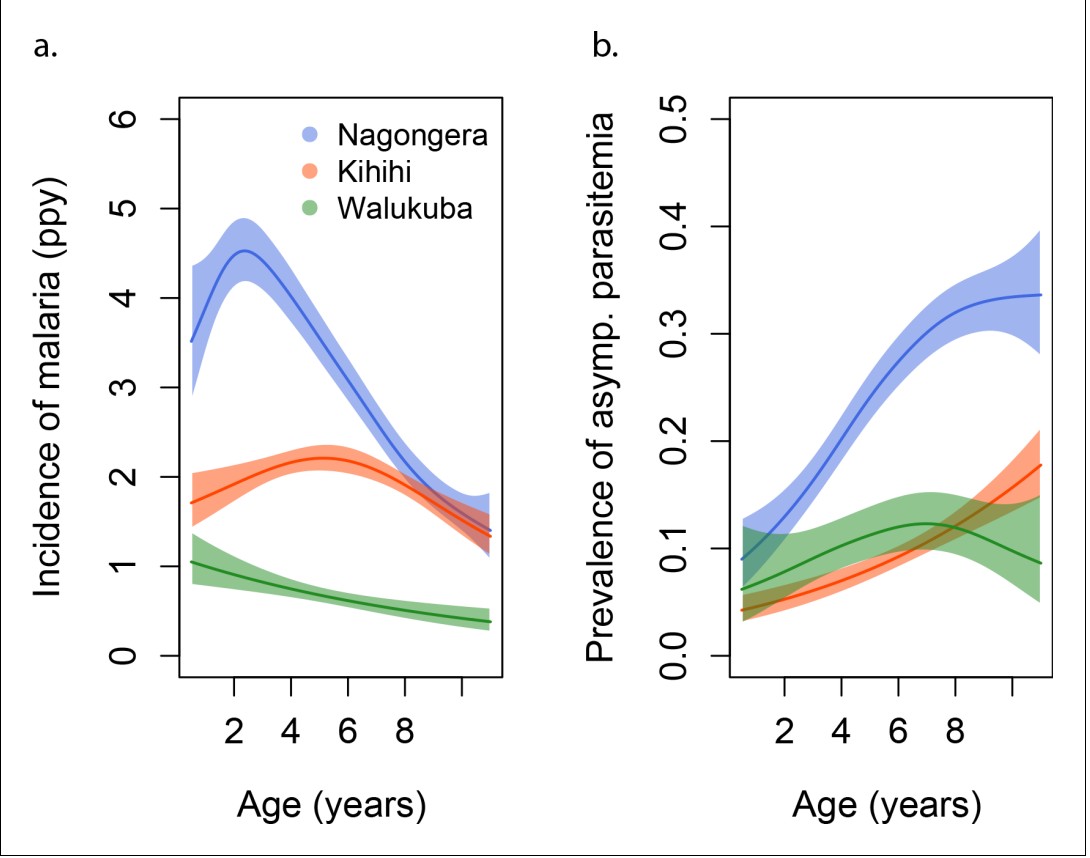

**Figure 1.** Incidence and prevalence of malaria as a function of age. (a) and prevalence of asymptomatic parasitemia (b) in the three study sites as a function of age, modeled using generalized additive models (GAMS). Shaded areas represent 95% confidence bounds.

DOI: https://doi.org/10.7554/eLife.35832.004

We considered multiple candidate models to describe the association between parasite density, age and aEIR (Appendix 1). Models allowing smooth (non-linear) relationships with aEIR best fit the data. Models allowing for two-way interactions between age and aEIR also outperformed models that did not include interactions.

In moderate and high transmission settings (households with aEIR >5), increasing age and increasing exposure were independently and linearly associated with decreases in the parasite densities (*Table 2*). On average, parasite densities decreased by a factor of 0.76 (95%CI 0.75–0.77) for each additional year of age and by a factor of 0.73 (95%CI 0.70–0.76) for each two-fold increase in the aEIR. The relationship was less evident for the lower transmission households (aEIR <5). In these settings, there continued to be a decreasing (although smaller) association with age, but the expected parasite densities at any given age were equal or lower to those observed in the higher exposure (aEIR >10) settings.

*Figures 4a* and *5a* present the predicted parasite densities, as a function of age and aEIR, according to the best fitting model. While an individual aged 1 year exposed to an aEIR of 10 is expected to develop a parasite density of 14,610 parasites/μL (95% CI 5924–36,031 parasites/μL) upon infection, the expected parasite density goes down to 3237 parasites/μL (95% CI 1381–7586 parasites/μL) by age 10 years. In contrast, the expected parasite density in an individual living in a setting with aEIR of 150 will be similar at age 1 year (13,071 parasites/μL (95% CI 5256–32,503 parasites/μL)), but significantly lower by age 10 years (999 parasites/μL (95% CI 398–2508 parasites/μL)).

To test whether the observed associations with age could be explained by the cumulative exposure over a life time, we also fit models where, instead of adjusting for the aEIR, we adjusted for the cumulative number of infectious bites (i.e. the product of age and aEIR) (*Figure 5—figure*

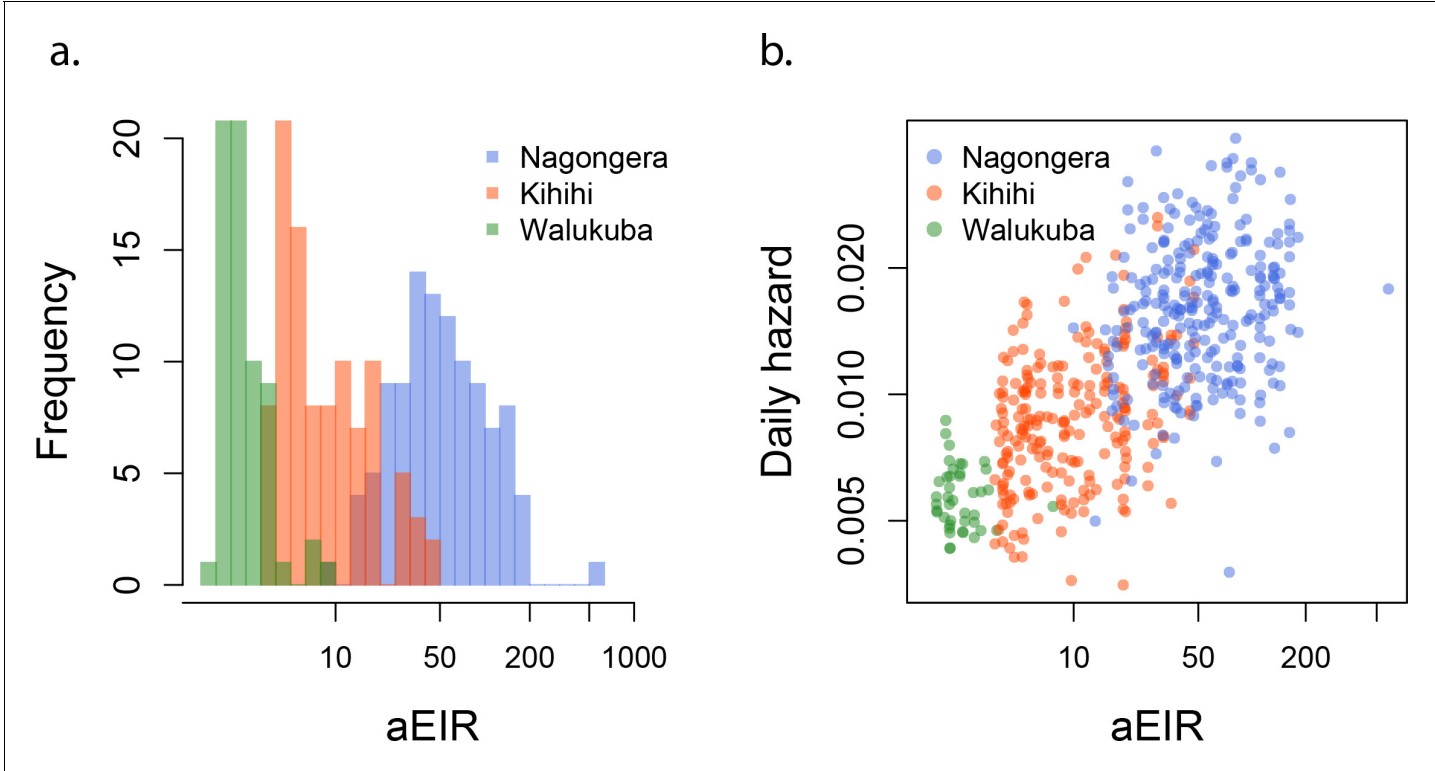

**Figure 2.** aEIR as a metric of individual exposure. (a) Distribution of the average annual entomological inoculation rate (aEIR) experienced by the study households in the three study sites. (b) Correlation between the measured aEIRs and the estimated individual hazards of infection.
DOI: https://doi.org/10.7554/eLife.35832.005

*supplement 2*). Results from these models are consistent with a smaller, yet independent effect of age on the development of anti-parasite immunity; for any given level of cumulative exposure, each additional year of life was associated with decreases in parasite densities by a factor of 0.82 (95%CI 0.81–0.85).

## Anti-disease immunity

We define anti-disease immunity as the ability to tolerate a given parasite density without developing objective fever. Thus, we were interested in modeling temperatures recorded at specific parasite densities, as a function of age and aEIR. Consistent with models characterizing anti-parasite immunity, models including smooth effects and interactions fitted the data significantly better than simpler models.

As expected, we found a strong association between parasite densities and objective temperature (*Figure 6—figure supplement 1*). Increases in parasite densities above 1000 parasites/μL were associated with higher expected temperatures across ages and transmission settings. In addition, we found a negative association between objective temperature at a given parasite density and age (*Figures 3b*, *4b* and *6*). In moderate and high transmission settings (aEIR >5), the objective temperature at a given parasite density decreased on average by 0.08°C (95% CI 0.07–0.10°C) for each additional year of life (*Table 2*). Thus, while the expected temperature for a child aged 1 year living in a setting with aEIR of 10 with a parasite density of 40,000 would be 38.8°C (95% CI 38.5–39.2°C), the expected temperature would decrease to 37.6°C (95% CI 37.3–38.0°C) if the same child experienced the infection at age 10 years (*Figures 4b* and *6*). This association was similar even when adjusting for cumulative exposure and for the differences in incidence of non-malarial fever across age-groups (*Figure 5—figure supplement 5*).

Similar to the anti-parasite immunity results described above, the observed association between exposure level and anti-disease immunity was less evident than the association with age (*Figures 3b*, *4b* and *6*). For moderate and high transmission settings (aEIR 5 to 300), there was a linear negative

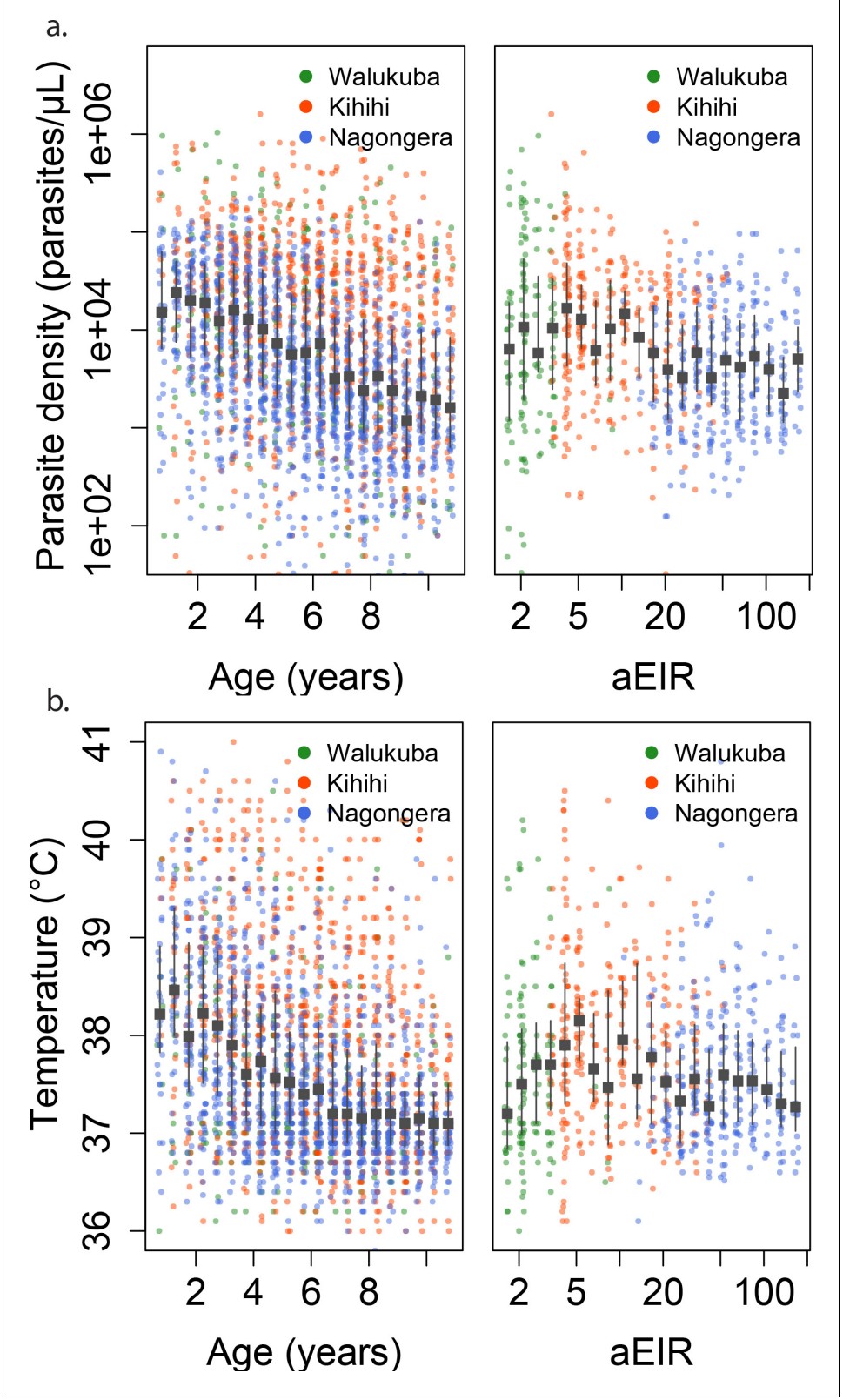

**Figure 3.** Changes in parasite densities and objective temprature as a function of age and exposure. Trends in parasite densities. (a) recorded during symptomatic (passive surveillance) infections and routine (active surveillance) visits as a function of age (left) and aEIR (right); and trends in the objective temperature (b) recorded during visits in which participants were found to have a parasite density between 50,000 parasites/μL and 200,000

*Figure 3 continued on next page*

*Figure 3 continued*

parasites/µL, as a function of age (left) and aEIR (right). Each point represents a measurement obtained during a study visit. The median and interquartile range are shown in black.

DOI: https://doi.org/10.7554/eLife.35832.006

association between objective temperature at a given parasite density and aEIR. The objective temperature decreased by 0.07°C (95% CI 0.05–0.10°C) for each two-fold increase in aEIR. However, the relationship did not follow this trend for lower transmission settings (*Table 2*). Children living in the lowest transmission settings (aEIR 1 to 5) appeared to tolerate higher parasite densities than children living in moderate transmission settings (aEIR 5 to 10).

As an alternative way to characterize anti-disease immunity, we used our best fitting model to predict the fever threshold, defined as the minimum parasite density associated with objective fever (temperature >38°C), across levels of age and aEIR (*Figure 5b*). This quantity is often referred to as the 'pyrogenic density'. Results from this analysis show that, for settings with moderate and high transmission (aEIR >5), the fever threshold increases both with age and increasing exposure. Thus, while a 1-year-old child living in a setting with aEIR of 10 presenting with a parasite density as low as 3747 parasites/µL (95% CI 777–11,129 parasites/µL) will be expected to be febrile, children older than 6 years of age exposed to very high transmission (aEIR 150) might be afebrile even with parasite densities higher than 60,000 parasites/µL.

## Overall immunity against symptomatic malaria

Finally, to characterize the association between age and aEIR on the overall risk of developing symptomatic malaria upon infection (i.e. the combined effect of anti-parasite and anti-disease immunity), we fit a series of models where the outcome of each independent microscopically detectable infection (i.e. symptomatic malaria or asymptomatic parasitemia) was modeled as a function of age and aEIR. Models allowing smooth relationships, with or without two-way interactions, fit the data equally well.

Results from this analysis are consistent with results from the anti-parasite and anti-disease models (*Figure 7*). While young children living in low transmission settings (aEIR = 5) develop symptomatic malaria in most their infections, the probability that an infection results in symptomatic malaria decreases as a function of age and exposure. The expected probability of symptomatic disease for a child aged 1 year living in a setting with aEIR of 50 is 0.92 (95% CI 0.79–0.97), but it decreases to 0.51 (95% CI 0.29–0.73) by age 10 years.

## Impact of recent infection on immunity

To assess whether recent *P. falciparum* infection was associated with different levels of anti-parasite and anti-disease immunity, we used data on the recent malaria history of each individual to fit

**Table 2.** Results of linear models quantifying the association between age, aEIR and immunity outcomes.

|  | All data | aEIR $\geq$ 5 (n = 5047) | aEIR < 5 (n = 593) |
|---|---|---|---|
| Anti-parasite immunity | Fold change in parasite density (95% CI) | | |
| Age (*years*) | 0.78 (0.77, 0.79) | 0.76 (0.75, 0.77) | 0.87 (0.83, 0.90) |
| Log$_2$ aEIR | 0.82 (0.79, 0.84) | 0.73 (0.69, 0.77) | 1.92 (1.69, 2.15) |
| Anti-disease immunity* | Change in objective temperature C (95% CI) | | |
| Age (*years*) | −0.07 (−0.06, −0.08) | −0.08 (−0.07, −0.1) | −0.04 (−0.07, −0.01) |
| Log$_2$ aEIR | −0.02 (−0.04, 0.0) | −0.07 (−0.05, −0.1) | 0.27 (0.11, 0.44) |
| Overall immunity against symptomatic malaria | Odds ratio of symptomatic disease (95% CI) | | |
| Age (*years*) | 0.78 (0.75, 0.82) | 0.77 (0.74, 0.80) | 0.90 (0.83, 0.99) |
| Log$_2$ aEIR | 0.91 (0.74, 1.13) | 0.62 (0.48, 0.80) | 3.83 (1.39, 10.6) |

*Model adjusted as well for Log parasite density.

DOI: https://doi.org/10.7554/eLife.35832.007

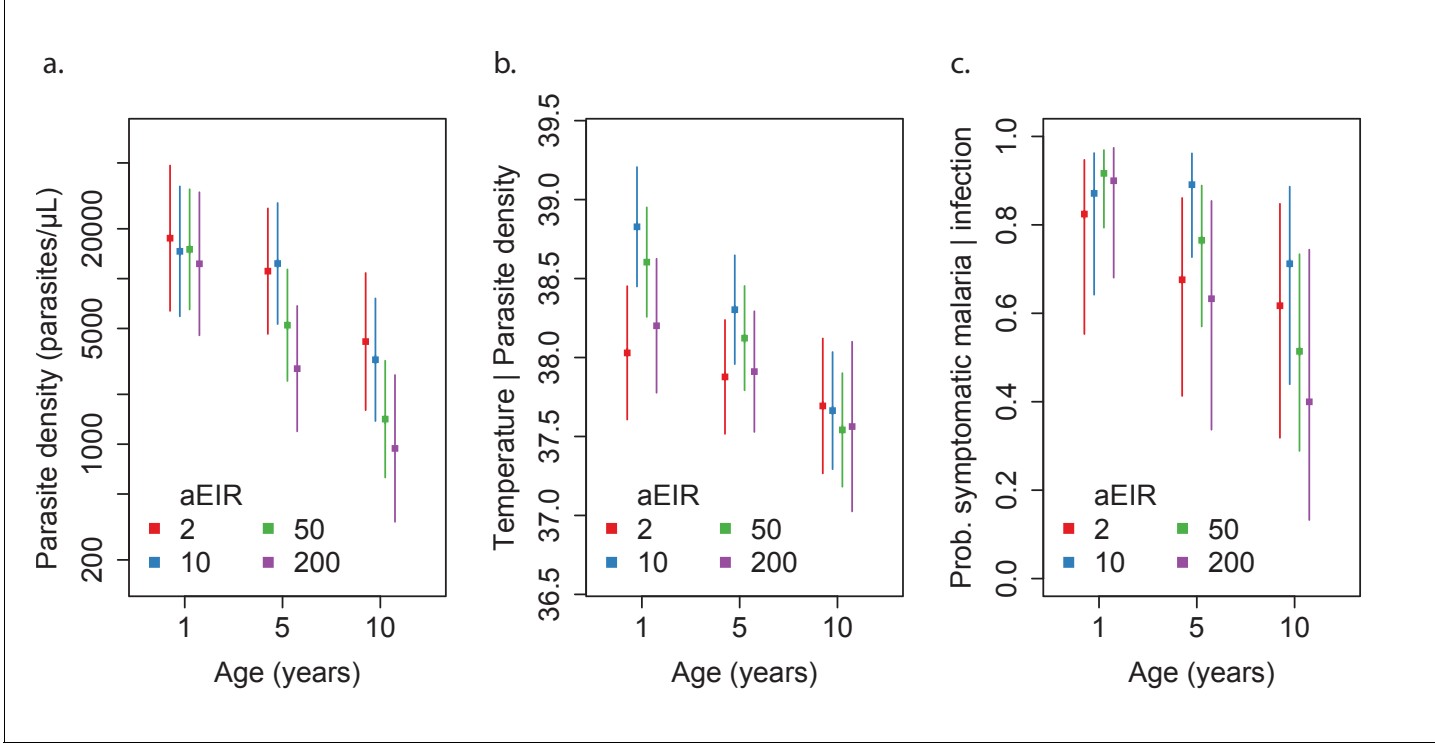

**Figure 4.** Results of best fitting models quantifying immunity. (**a**), anti-disease immunity (**b**) and overall immunity against symptomatic malaria (**c**). Each plot shows, for specific ages and aEIRs, the expected parasite density (/μL) (**a**), objective temperature given a density of 40,000 parasites/μL (**b**) and the probability of developing symptomatic malaria upon infection (**c**), estimated using the best fitting model. 95% confidence intervals of the estimates are also shown.

DOI: https://doi.org/10.7554/eLife.35832.008

models adjusted for number of *P.falciparum* positive visits in the past 3 and 6 months. We found no association between the number of recent malaria infections and our outcomes of interest (Appendix 2, *figure 5—figure supplement 3*).

## Development of anti-parasite and anti-disease immunity at the individual level

Models that included random effects at the individual and household levels outperformed models that assumed independence of observations, consistent with large heterogeneity between individuals in the development of anti-parasite, anti-disease and overall immunity against symptomatic malaria. To illustrate this heterogeneity, we used the best fitting model to predict the trajectories of a subset of individuals with respect to anti-parasite and anti-disease immunity, as a function of age and aEIR (*Figure 5—figure supplement 12*).

## Sensitivity analyses

Our main analyses include data from all visits regardless of their type (routine vs passive case detection). Thus, the expected values modeled here may be biased by the frequency of active vs passive episodes detected. In particular, it is possible that we have under-sampled the instances of asymptomatic infection, and thus, our estimates of the expected parasite densities may be an over-estimate of those present in the population. Similarly, it is also possible that consecutive asymptomatic infections represent persistent, rather than new infections. To address these limitations, we performed sensitivity analyses where we (a) up-weighted the episodes of asymptomatic parasitemia, to account for potentially unobserved asymptomatic infections and (b) included only 'incident' asymptomatic infections, under the assumption that subsequent asymptomatic samples represented persistent (rather than new) infections. Results from these analyses were qualitatively identical to the main

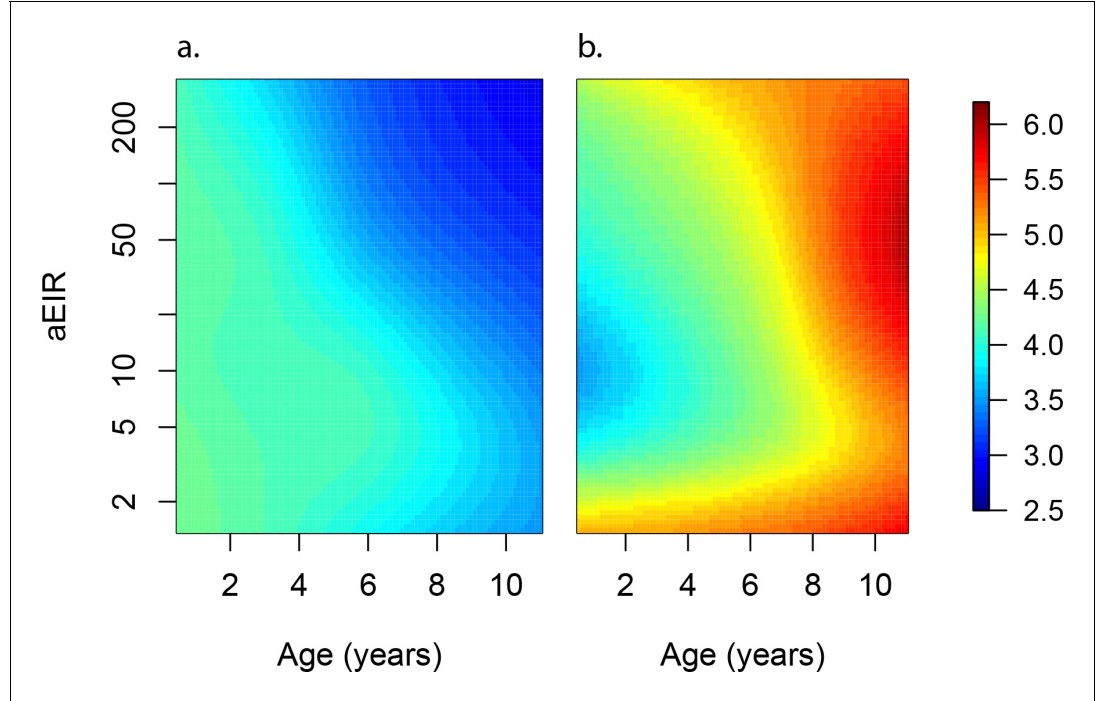

**Figure 5.** Anti-parasite and Anti-disease immunity as a function of age and exposure. (**a**) and anti-disease immunity (**b**). These results are similar to those presented in *Figure 4*, but for the full range of ages and aEIRs included in the data. Panel (**a**) shows expected parasite densities (parasites/µL, log 10) upon infection for different ages and levels of exposure (aEIR). Panel (**b**) shows the 'fever threshold' or 'pyrogenic density', the minimum parasite densities (parasites/µL, log 10) associated with fever (temperature 38°C or greater), again as a function of age and exposure.

DOI: https://doi.org/10.7554/eLife.35832.009

The following figure supplements are available for figure 5:

**Figure supplement 1.** Confidence bounds of best fitting model quantifying anti-parasite immunity.
DOI: https://doi.org/10.7554/eLife.35832.010

**Figure supplement 2.** Adjusting for cumulative exposure.
DOI: https://doi.org/10.7554/eLife.35832.011

**Figure supplement 3.** Adjusting for the number of infections in the past 3 months: Results of models.
DOI: https://doi.org/10.7554/eLife.35832.012

**Figure supplement 4.** Adjusting for the probability of non-malaria fevers.
DOI: https://doi.org/10.7554/eLife.35832.013

**Figure supplement 5.** Adjusting for the probability of non-malaria fevers.
DOI: https://doi.org/10.7554/eLife.35832.014

**Figure supplement 6.** Adjusting for the probability of observation.
DOI: https://doi.org/10.7554/eLife.35832.015

**Figure supplement 7.** Limiting the analysis to 'incident' infections.
DOI: https://doi.org/10.7554/eLife.35832.016

**Figure supplement 8.** Limiting the analysis to individuals without the sickle hemoglobin mutation (β globin E6V).
DOI: https://doi.org/10.7554/eLife.35832.017

**Figure supplement 9.** Limiting the analysis to individuals from Tororo and Kanungu.
DOI: https://doi.org/10.7554/eLife.35832.018

**Figure supplement 10.** Limiting the analysis to individuals living in settings with aEIR $\geq$5.
DOI: https://doi.org/10.7554/eLife.35832.019

**Figure supplement 11.** Using a metric of aEIR that only includes prior observations.
DOI: https://doi.org/10.7554/eLife.35832.020

**Figure supplement 12.** Predicted individual trajectories of anti-disease and anti-parasite immunity.
DOI: https://doi.org/10.7554/eLife.35832.021

**Figure supplement 13.** Model checks.
DOI: https://doi.org/10.7554/eLife.35832.022

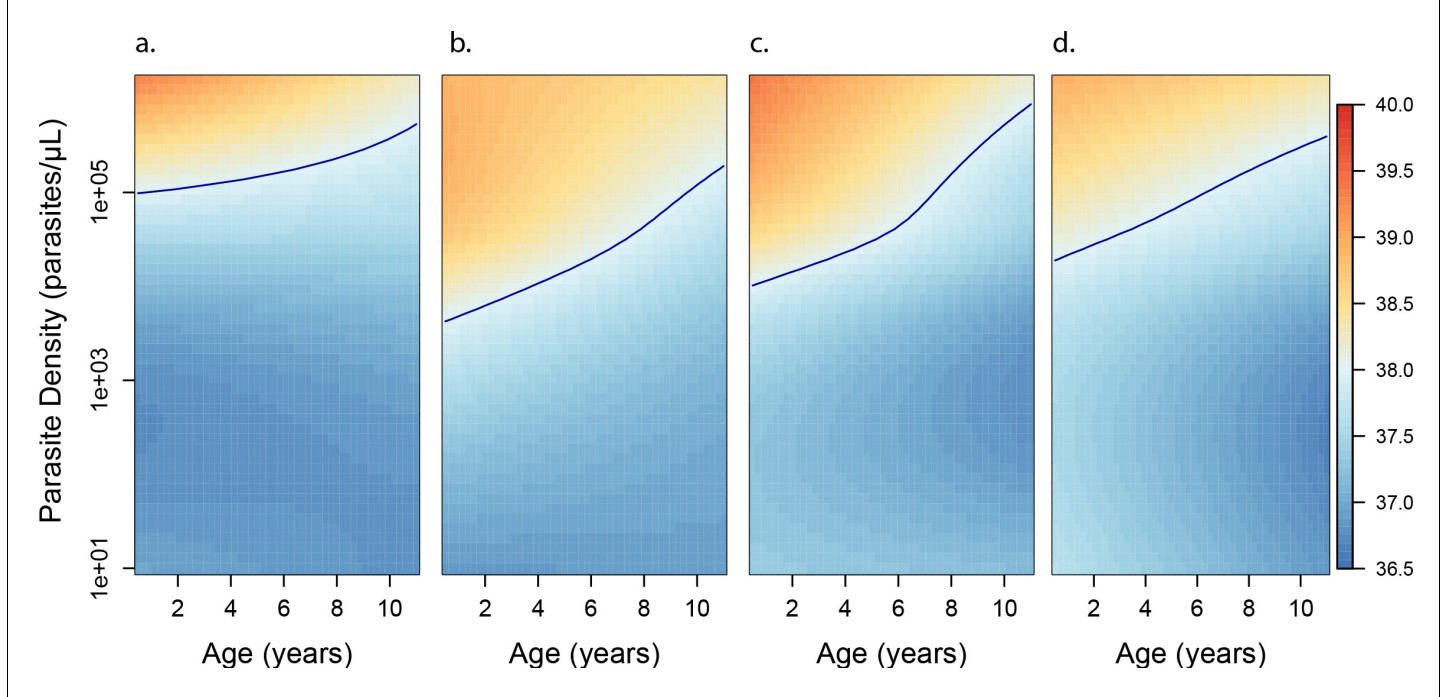

**Figure 6.** Anti-disease immunity as a function of age and exposure. Each panel shows how the expected objective temperature (°C) varies as a function of age and parasite density, for different transmission settings. (**a**) aEIR = 2; (**b**) aEIR = 10; (**c**) aEIR = 50; (**d**) aEIR = 200. Contours indicating the fever threshold (38°C) are also shown. Confidence bounds for these plots are presented in *Figure 6—figure supplement 1*.

DOI: https://doi.org/10.7554/eLife.35832.023

The following figure supplements are available for figure 6:

**Figure supplement 1.** Association between parasite density (parasites/μL) and objective temperature (in °C).

DOI: https://doi.org/10.7554/eLife.35832.024

**Figure supplement 2.** Confidence bounds of best fitting model quantifying anti-disease immunity.

DOI: https://doi.org/10.7554/eLife.35832.025

analysis reported here and are presented in the supplementary material (*Figure 5—figure supplements 6* and *7*).

To explore whether differences in the prevalence of certain host genetic polymorphisms between sites could be driving some of our findings, we also performed sensitivity analyses limiting the dataset to those subjects without the sickle hemoglobin mutation (β globin E6V), known to protect against malaria (*Lopera-Mesa et al., 2015*; *Taylor et al., 2012*). Even though the sample size of these analyses was smaller (observations from 155/773 individuals were excluded), results were unchanged qualitatively (*Figure 5—figure supplement 8*). Similarly, restricting the dataset to children without two other known polymorphisms (the α-thalassemia 3.7 kb deletion or glucose-6-phosphate dehydrogenase deficiency caused by the common African variant (G6PD A-)), had little effect on the results.

## Discussion

Our findings illustrate how anti-parasite and anti-disease immunity develop gradually and in parallel, complementing each other in reducing the probability of experiencing symptomatic disease upon infection with *P. falciparum*. While anti-parasite immunity acts to restrict the parasite densities that develop upon each subsequent infection, anti-disease immunity increases the tolerance to high parasite densities. Thus, older children are less likely to develop symptomatic malaria upon infection both because they tolerate parasite densities better without developing fever, and because they are less likely to develop high parasite densities.

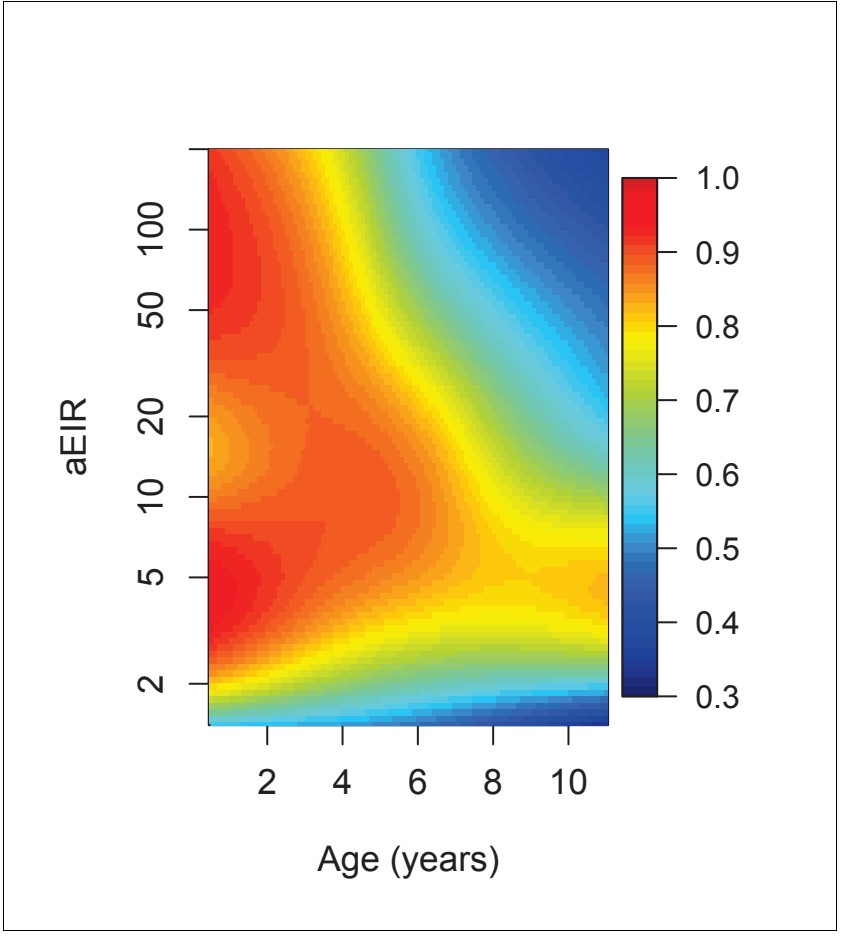

**Figure 7.** Results of the best model quantifying overall immunity against symptomatic malaria (upon microscopically detectable infection), as a function of age and exposure (aEIR). Colors represent the expected probability of developing symptomatic malaria upon infection. Confidence bounds for these plots are presented in *Figure 7—figure supplement 1*.

DOI: https://doi.org/10.7554/eLife.35832.026

The following figure supplement is available for figure 7:

**Figure supplement 1.** Confidence bounds of best fitting model quantifying overall immunity against symptomatic malaria.

DOI: https://doi.org/10.7554/eLife.35832.027

Our results indicate independent effects of age on the acquisition of both anti-parasite and anti-disease immunity. These independent age effects may reflect maturation of the immune system as well as other physiological changes that decrease the propensity to fever (*Struik and Riley, 2004*; *Baird, 1998*). Furthermore, our findings are consistent with independent effects of transmission intensity on the acquisition of these two types of immunity. While the results obtained for moderate and high transmission settings (aEIR >5) are consistent and expected, and suggest that immunity develops faster in settings where individuals get infected by *P. falciparum* more often, the results obtained for the lowest transmission settings are harder to reconcile. These results were largely driven by observations collected in the Walukuba site, and as such it is possible that site-specific characteristics may have driven them. Walukuba was previously a relatively high transmission rural area, but substantial decreases in transmission intensity have been observed since 2011, likely due to urbanization. While our sensitivity analyses suggested that differences in the prevalence of three well characterized host-genetic polymorphisms between sites do not explain these discrepant results, it is still possible that other unmeasured site-specific characteristics may have driven them. Lower complexity of infection coupled with lower parasite diversity in Walukuba, for example, could

cause this difference, as developing an effective immune response against fewer parasite strains may be much easier than developing immunity against a more diverse pool (*Hviid, 1998*; *Bull et al., 1998*). Testing this hypothesis would require careful characterization of the complexity and diversity of infections in each of our cohort settings.

While site-specific characteristics may underlie the observed high levels of clinical immunity against malaria in the low transmission setting, it is also possible that this finding reflects biologically relevant differences in how immunity against malaria develops. For example, it has been hypothesized that immunity may develop optimally in individuals that are exposed at a low rate, and that more frequent infections may interfere with the development of robust immune responses (*Wipasa et al., 2010*; *Langhorne et al., 2008*). Answering this question will require further detailed studies across transmission settings, with careful characterization of both exposure and infection outcomes.

There are several limitations to this study. With a study design including routine visits every 3 months, we are likely to have missed several asymptomatic infections, particularly in the moderate and high transmission settings. Moreover, since infections were detected using microscopy, we were unable to detect sub-patent infections, and we lack knowledge about the genetic complexity of each infection. While it is possible that the expected values modeled here (expected parasite density and fever threshold) were biased by these sources of measurement error, sensitivity analyses suggest that the relationships observed were robust. Secondly, while we found an independent association between the average household aEIR and both anti-parasite and anti-disease immunity, it is not clear that this is the most relevant metric of exposure for the development of clinical immunity to malaria. Alternative metrics such as the number of discrete infections, the number of 'strains' seen or the total parasite-positive time might be more relevant, but require the collection of additional data, including more frequent sampling. Finally, while this study provides very detailed insight into how two types of clinical immunity to malaria develop in endemic settings as a function of age and repeated exposure, it says nothing about the duration of immunity.

Prior studies have tried to model the processes driving acquisition of clinical immunity against malaria. However, these models have been generally informed by aggregated epidemiological data (age-incidence and age-prevalence) which limits their capacity to isolate the contributions of age and repeated exposure (*Filipe et al., 2007*; *Griffin et al., 2015*, *2014*). Our results quantify how anti-parasite and anti-disease immunity develop in children across the malaria transmission spectrum, and they support strong independent effect of age and a perhaps paradoxical effect of exposure. The methods proposed here to model anti-parasite and anti-disease immunity may also provide a framework to select individuals with immune and non-immune phenotypes for evaluations of immune correlates of protection.

## Materials and methods

### Ethics statement

The study protocol was reviewed and approved by the Makerere University School of Medicine Research and Ethics Committee (Identification numbers 2011–149 and 2011–167, the London School of Hygiene and Tropical Medicine Ethics Committee (Identification numbers 5943 and 5944), the Durham University School of Biological and Biomedical Sciences Ethics Committee (PRISM Entomology Uganda), the University of California, San Francisco, Committee on Human Research (Identification numbers 11–05539 and 11–05995) and the Uganda National Council for Science and Technology (Identification numbers HS-978 and HS-1019). All parents/guardians were asked to provide written informed consent at the time of enrollment.

### Data

We used data from three parallel cohort studies conducted in Uganda in sub-counties chosen to represent varied malaria transmission (*Kamya et al., 2015*). Walukuba, in Jinja district, is a peri-urban area near Lake Victoria that has the lowest transmission among the three (annual entomological inoculation rate (aEIR) estimated to be 2.8 [*Kamya et al., 2015*]). Kihihi, in Kanungu district, is a rural area in southwestern Uganda characterized by moderate transmission (aEIR = 32). Nagongera, Tororo district, is a rural area in southeastern Uganda with the highest transmission

(aEIR = 310) (*Kamya et al., 2015*; *Kilama et al., 2014*). Details on how the study households and participants were selected has been described elsewhere (*Kamya et al., 2015*). Briefly, all households were enumerated, and then approximately 100 households were selected at random from each site. Between August and September 2011, all children from these households aged between 6 months and 10 years who met eligibility criteria were invited to participate. As the cohorts were dynamic, additional children from participating households were invited to participate if they became eligible while the study was ongoing. Unless participants were withdrawn from the study either voluntarily or because they failed to comply with study visits, they were followed-up until they reached 11 years of age. Children from 31 randomly selected additional households were enrolled between August and October 2013 to replace households in which all study participants had been withdrawn. For this analysis, we used data collected from visits between August 2011 and November 2014.

The studies included passive and active follow-up of participants. Parents/guardians were encouraged to bring their children to designated study clinics for any illness. All medical care was provided free of charge, and participants were reimbursed for transportation costs. All children who reported fever in the previous 24 hr or were febrile at the time of the visit (tympanic temperature >38.0°C) were tested for malaria infection with a thick blood smear. Light microscopy was performed by an experienced laboratory technician who was not involved in direct patient care and verified by a second technician. Parasite density was calculated by counting the number of asexual parasites per 200 leukocytes (or per 500 leukocytes, if the count was <10 asexual parasites/200 leukocytes), assuming a leukocyte count of 8,000/µl. A blood smear was considered negative when no asexual parasites were found after examination of 100 high-power fields.

If the smear was positive, the patient was diagnosed with symptomatic malaria and received treatment with artemether-lumefantrine (AL), the recommended first-line treatment in Uganda. Episodes of complicated or recurrent malaria occurring within 14 days of therapy were treated with quinine. In addition, routine evaluations were performed every 3 months, including testing for asymptomatic parasitemia using thick blood smears.

Entomological surveys were also conducted every month at all study households. During these surveys, mosquitoes were collected using miniature CDC light traps (Model 512; John W. Hock Company). Established taxonomic keys were used to identify female *Anopheles* mosquitoes. Individual mosquitoes were tested for sporozoites using an ELISA technique (*Kilama et al., 2014*). All female *Anopheles* mosquitoes captured in Walukuba and Kihihi were tested; in Nagongera testing was limited to 50 randomly selected female *Anopheles* mosquitoes per household per night due to the large numbers collected. Therefore, for each household and/or site it was possible to calculate multiple entomological metrics, including the average human biting rate (average number of *female Anopheles* mosquitoes caught in a household per day), the average sporozoite rate (the average proportion of mosquitos that tested positive for *Plasmodium falciparum*) and the entomological inoculation rate (EIR, the product of the household human biting rate and the site sporozoite rate).

## Statistical analyses

The purpose of these analyses was to model and quantify the development of immunity against symptomatic malaria, as a function of age and exposure, measured by the household EIR.

We modeled two specific types of immunity that have been previously described as components of immunity to malaria. We defined anti-parasite immunity as the ability to control parasite densities upon infection and anti-disease immunity as the ability to tolerate parasite infections without developing objective fever. Thus, for models of anti-parasite immunity, the outcome of interest was the parasite density recorded (using thick blood smear) at each parasite-positive study visit. For models of anti-disease immunity, the outcome of interest was the objective temperature recorded during parasite positive visits, conditional on the parasite density. In addition, we also modeled overall immunity against symptomatic malaria. For these analyses, the outcome of interest was the probability of presenting with fever given infection (parasite positivity).

In order to model the association between the outcomes and covariates of interest we used generalized additive models (gams). Gams provide a good framework, as they allow for smooth non-linear relationships. Details on the specific models explored are provided in the supplementary material (Appendix 1). In summary, the models followed the following form.

(1) Anti-parasite immunity

$$Log_{10}(Parasite\ density)_{ijk} = f\left(age_{ijk},\ Log_2 aEIR_j\right) + u_i + \gamma_j$$

(2) Anti-disease immunity

$$Temperature_{ijk} = f\left(age_{ijk},\ Log_2 aEIR_j,\ Log_{10} Parasite\ density_{ijk}\right) + u_i + \gamma_j$$

(3) Overall immunity against symptomatic malaria

$$P(symptomatic\ malaria\ upon\ infection)_{ijk} = f\left(age_{ijk},\ Log_2 aEIR_j\right) + u_i + \gamma_j$$

where $i$ is an index for individuals, $j$ for households and $k$ for specific visits. Thus, $age_{ijk}$ represents the age of child $i$ from household $j$ during visit $k$, and $aEIR_j$ represents the average annual EIR recorded for household $j$. We included the EIR as an average (time-invariant) covariate, as we were interested in modeling the impact of the average exposure to malaria over time on the development of clinical immunity. Therefore, our model implicitly assumes that malaria transmission has been relatively stable at these three sites. To account for lack of independence, all models included random effects at the individual ($u_i$) and household ($\gamma_i$) levels.

All our primary analyses included the full dataset. However, since results were consistent with a non-monotonic relationship between aEIR and the outcomes of interest, we also fit models stratified by aEIR (aEIR $\geq$5 vs. aEIR <5). All models were fitted in the R statistical framework using package mgcv (*R Core Team, 2016*). Best fitting models were selected based on Akaike's Informaiton Criterion, but changes in the percent deviance explained are also presented.

## Code and data availability

All the data used for these analyses as well as the R code used to reproduce the main study findings are available at https://github.com/isabelrodbar/immunity (*Rodriguez-Barraquer, 2018*; (copy archived at https://github.com/elifesciences-publications/immunity). Complete data from the 3 cohort studies are available in the ClinEpiDB website (https://clinepidb.org/ce/app).

Confidence bounds are presented in *Figure 5—figure supplement 1*.

## Acknowledgements

We thank all study participants who participated in this study and their families. We also thank the study team and the Makerere University–UCSF Research Collaboration and the Infectious Diseases Research Collaboration for administrative and technical support.

## Additional information

### Funding

| Funder | Grant reference number | Author |
|---|---|---|
| National Institutes of Health | 2U19AI089674 | Isabel Rodriguez-Barraquer<br>Emmanuel Arinaitwe<br>Prasanna Jagannathan<br>Moses R Kamya<br>Phillip J Rosenthal<br>John Rek<br>Grant Dorsey<br>Joaniter Nankabirwa<br>Sarah G Staedke<br>Maxwell Kilama<br>Chris Drakeley<br>Isaac Ssewanyana<br>David L Smith<br>Bryan Greenhouse |
| Bill and Melinda Gates Foundation | OPP1110495 | David L Smith |

The funders had no role in study design, data collection and interpretation, or the decision to submit the work for publication.

## Author contributions
Isabel Rodriguez-Barraquer, Conceptualization, Data curation, Formal analysis, Investigation, Visualization, Methodology, Writing—original draft, Writing—review and editing; Emmanuel Arinaitwe, Joaniter Nankabirwa, Investigation, Project administration, Writing—review and editing; Prasanna Jagannathan, Formal analysis, Investigation, Writing—review and editing; Moses R Kamya, Supervision, Investigation, Methodology, Writing—review and editing; Phillip J Rosenthal, Grant Dorsey, Chris Drakeley, Conceptualization, Funding acquisition, Investigation, Writing—review and editing; John Rek, Sarah G Staedke, Supervision, Investigation, Writing—review and editing; Maxwell Kilama, Data curation, Investigation, Writing—review and editing; Isaac Ssewanyana, Investigation, Writing—review and editing; David L Smith, Conceptualization, Formal analysis, Writing—original draft; Bryan Greenhouse, Conceptualization, Formal analysis, Funding acquisition, Methodology, Writing—original draft, Writing—review and editing

## Author ORCIDs
Isabel Rodriguez-Barraquer http://orcid.org/0000-0001-6784-1021
Prasanna Jagannathan https://orcid.org/0000-0001-6305-758X
Chris Drakeley https://orcid.org/0000-0003-4863-075X
David L Smith https://orcid.org/0000-0003-4367-3849

## Ethics
Human subjects: The study protocol was reviewed and approved by the Makerere University School of Medicine Research and Ethics Committee (Identification numbers 2011-149 and 2011-167), the London School of Hygiene and Tropical Medicine Ethics Committee (Identification numbers 5943 and 5944), the Durham University School of Biological and Biomedical Sciences Ethics Committee (PRISM Entomology Uganda), the University of California, San Francisco, Committee on Human Research (Identification numbers 11-05539 and 11-05995) and the Uganda National Council for Science and Technology (Identification numbers HS350 and HS-1019). All parents/guardians were asked to provide written informed consent at the time of enrollment.

## Decision letter and Author response
Decision letter https://doi.org/10.7554/eLife.35832.044
Author response https://doi.org/10.7554/eLife.35832.045

# Additional files

## Supplementary files
• Transparent reporting form
DOI: https://doi.org/10.7554/eLife.35832.028

## Data availability
All the data used for these analyses as well as the R code used to reproduce the main study findings are available at https://github.com/isabelrodbar/immunity (copy archived at https://github.com/elifesciences-publications/immunity). Complete data from the 3 cohort studies are available at the CliEpiDB website (https://clinepidb.org/ce/app/record/dataset/DS_0ad509829e).

The following previously published dataset was used:

| Author(s) | Year | Dataset title | Dataset URL | Database, license, and accessibility information |
| --- | --- | --- | --- | --- |
| Grant Dorsey | 2017 | PRISM cohort study | https://clinepidb.org/ce/app/record/dataset/DS_0ad509829e | Publicly available at ClinEpiDB (https://clinepidb.org/ce/app). |

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

## Appendix 1

DOI: https://doi.org/10.7554/eLife.35832.029

## Detailed description of the models

### Anti-parasite immunity

### Models

To explore the association between age, exposure (aEIR) and parasite density upon infection, we fit the following models.

AP1: $Log_{10}(Parasite\ density)_{ijk} = age_{ijk} + Log_2aEIR_j + u_i + \gamma_j$

AP2: $Log_{10}(Parasite\ density)_{ijk} = f(age_{ijk}) + f(Log_2aEIR_j) + u_i + \gamma_j$

AP3: $Log_{10}(Parasite\ density)_{ijk} = age_{ijk} : aEIR_j + u_i + \gamma_j$

AP4: $Log_{10}(Parasite\ density)_{ijk} = f(age_{ijk},\ Log_2aEIR_j) + u_i + \gamma_j$

Where $i$ is an index for individuals, $j$ for households and $k$ for specific visits. Thus, $age_{ijk}$ represents the age of child $i$ from household $j$ during visit $k$, and $aEIR_j$ represents the average annual EIR recorded for household $j$. $u_i$ and $\gamma_j$ denote the individual and household random effects, respectively, assumed to be normally distributed with mean 0. Tensor interactions were used for interaction models (AP4).

**Appendix 1—table 1.** Anti-parasite immunity Model fit.

| Model | DF | % Deviance Explained | AIC |
| --- | --- | --- | --- |
| AP1 | 418.7 | 33.2 | 14052 |
| AP2 | 414.2 | 33.1 | 14048 |
| AP3 | 408 | 33 | 14046 |
| AP4 | 404 | 33 | 14032 |

DOI: https://doi.org/10.7554/eLife.35832.030

### Anti-disease immunity

### Models

To explore the association between age, exposure (aEIR), parasite density and objective temperature, we fit the following models

AD1: $Temperature_{ijk} = age_{ijk} + Log_2aEIR_j + Log_{10}Parasite\ density_{ijk} + u_i + \gamma_j$

AD2: $Temperature_{ijk} = f(age_{ijk}) + f(Log_2aEIR_j) + f(Log_{10}Parasite\ density_{ijk}) + u_i + \gamma_j$

AD3: $Temperature_{ijk} = age_{ijk} : aEIR_j : Log_{10}Parasite\ density_{ijk} + u_i + \gamma_j :$

AD4: $Temperature_{ijk} = f(age_{ijk},\ Log_2aEIR_j,\ Log_{10}Parasite\ density_{ijk}) + u_i + \gamma_j$

Where $i$ is an index for individuals, $j$ for households and $k$ for specific visits. Thus, $age_{ijk}$ represents the age of child $i$ from household $j$ during visit $k$, and $aEIR_j$ represents the average annual EIR recorded for household $j$. $u_i$ and $\gamma_j$ denote the individual and household random effects, respectively, assumed to be normally distributed with mean 0. Tensor interactions were used for interaction models (AD4).

### Fever threshold

In order to estimate the fever threshold, defined as the minimum parasite density associated with fever (temperature greater than 38°C) we first used the best fitting model (AD3) to predict the expected temperature across levels of parasite density, age and exposure (aEIR) (as shown in *Figure 6*). For each level of age and aEIR, we then extracted the minimum parasite density that predicted a temperature >38°C. This was possible since the association between parasite density and temperature is positive (*Figure 6—figure supplement 1*).

**Appendix 1—table 2.** Anti-disease immunity Model fit.

| Model | DF | % Deviance Explained | AIC |
|---|---|---|---|
| AD1 | 315 | 37.7 | 14946 |
| AD2 | 285 | 39.0 | 14774 |
| AD3 | 314 | 38.3 | 14894 |
| AD4 | 296 | 39.4 | 14758 |

DOI: https://doi.org/10.7554/eLife.35832.031

## Overall immunity against symptomatic malaria

### Models

To explore the association between age, exposure (aEIR), and the odds of symptomatic malaria upon infection, we fit the following models.

SM1: $Log\ Odds(symptomatic\ malaria)_{ijk} = age_{ijk} + Log_2aEIR_j + u_i + \gamma_j$

SM2: $Log\ Odds(symptomatic\ malaria)_{ijk} = f(age_{ijk}) + f(aEIR_j) + u_i + \gamma_j$

SM3: $Log\ Odds(symptomatic\ malaria)_{ijk} = age_{ijk} : Log_2aEIR_j + u_i + \gamma_j$

SM4: $Log\ Odds(symptomatic\ malaria)_{ijk} = f(age_{ijk},\ Log_2aEIR_j) + u_i + \gamma_j$

Where $i$ is an index for individuals, $j$ for households and $k$ for specific visits. Thus, $age_{ijk}$ represents the age of child i from household j during visit k, and $aEIR_j$ represents the average annual EIR recorded for household $j$. $u_i$ and $\gamma_j$ denote the individual and household random effects, respectively, assumed to be normally distributed with mean 0. Tensor interactions were used for interaction models (SM4).

**Appendix 1—table 3.** Overall immunity against symptomatic malaria Model fit.

| Model | DF | % Deviance Explained | AIC |
|---|---|---|---|
| SM1 | 399.8 | 28 | 5382 |
| SM2 | 385.3 | 28 | 5373 |
| SM3 | 385.5 | 27.8 | 5380 |
| SM4 | 362.6 | 27.3 | 5369 |

DOI: https://doi.org/10.7554/eLife.35832.032

## Appendix 2

DOI: https://doi.org/10.7554/eLife.35832.033

### Models adjusted for recent exposure

To assess whether recent exposure to malaria was associated with different levels of anti-parasite and anti-disease immunity we fit models adjusted for number of P.*falciparum* positive visits in different windows of time (3 and 6 months). We fit models where the association was assumed to be linear, as well as models that allowed for a smooth relationship (for 6 months only). Data from the visits that occurred in the first 3 and 6 months since enrollment were excluded from these analyses, respectively. See *Figure 5—figure supplement 3*.

**Appendix 2—table 1.** Anti-parasite immunity.

| Time window | Functional form | Coefficient (95% CI) | p-value |
|---|---|---|---|
| 3 months | Linear | −0.003 (−0.03–0.03) | 0.83 |
| 6 months | Linear | 0.003 (−0.02–0.02) | 0.75 |
| 6 months | Smooth | - | 0.77 |

DOI: https://doi.org/10.7554/eLife.35832.034

**Appendix 2—table 2.** Anti-disease immunity.

| Time window | Functional form | Coefficient (95% CI) | p-value |
|---|---|---|---|
| 3 months | Linear | −0.01 (−0.04–0.02) | 0.38 |
| 6 months | Linear | −0.01 (−0.03–0.01) | 0.15 |
| 6 months | Smooth | - | 0.15 |

DOI: https://doi.org/10.7554/eLife.35832.035

