## [Decision Letter]

[Editors’ note: a previous version of this study was rejected after peer review, but the authors submitted for reconsideration. The first decision letter after peer review is shown below.]

Thank you for submitting your work entitled "Quantification of anti-parasite and anti-disease immunity to malaria as a function of age and exposure" for consideration by *eLife*. Your article has been reviewed by three peer reviewers, including Ben Cooper as the Reviewing Editor and Reviewer #1, and the evaluation has been overseen by a Senior Editor.

Our decision has been reached after consultation between the reviewers. Based on these discussions and the individual reviews below, we regret to inform you that your work will not be considered further for publication in *eLife*.

All reviewers thought the work presented some very detailed and interesting data. However, following consultation, the consensus was that by failing to account for individual exposure history in the data the work fell short of what was possible with these data, and also made implicit independence assumptions that seem hard to justify. If these limitations could be addressed in future work, the journal would be interested in considering a new manuscript looking at the same data.

Reviewer #1:

This manuscript presents an elegant application of generalised additive models to a rich longitudinal malaria data-set from 773 children across three study sites in Uganda. The authors use this statistical modelling approach to investigate how both anti-parasite immunity to *P. falciparum* and anti-disease immunity change with age and transmission intensity.

As the authors argue, this works represents a substantial advance over previous attempt to model processes driving malaria immunity which have been informed by aggregated data. The individual level data makes it possible to disentangle the effects of age and exposure history, and demonstrate an independent effect of age (i.e. beyond that which would be expected if exposure alone were driving acquisition of immunity).

Generally the manuscript is very well written, easy to read, clearly presented, and the findings accompanied with appropriate caveats. This work seems to make an important and worthwhile contribution to the literature. The sensitivity analysis in the supplementary material also considerably strengthens the work.

Reviewer #2:

This study analyses data from three longitudinal cohort studies from regions of Uganda with varying transmission intensity. The analysis is based on very detailed epidemiological data sets with data collected on *P. falciparum* parasite density, temperature, clinical episodes of malaria and EIR. In order to assess the statistical methods it is worthwhile to consider this analysis in the context of a previous analysis of Ugandan data by the same authors in an excellent paper published last year in The Journal of Infectious Diseases. The data utilised in this analysis is superior to that utilised in the previous paper in a number of ways: more participants; household measurements of EIR; individual level measurements of temperature and parasite density, etc. The absence of data on low density PCR-detectable infections however is a limitation. Although the statistical models utilised in this analysis are justifiable and robust, I feel that the methods have taken a step back compared to their previous methods.

In their previous analysis the authors captured how previous exposure to malaria affects the probability that an infection will lead to a clinical episode (e.g. individuals with a greater number of episodes in the past have greater levels of clinical immunity and thus a lower probability that a new infection progresses to an episode of clinical malaria). It appears to me that in this analysis that the rich data on an individuals' exposure history is not accounted for – and with a mean of 2 years' follow-up (over 3 years in some case) this is quite substantial data. Instead, the authors' models for temperature, parasite density and symptomatic malaria for visit k, depend only on measurements at visit k, and not on the individuals' malaria history at previous visits. I would like to see the authors utilise a method that accounts for malaria history at visits before k (e.g. such as the method used in their previous analysis).

Reviewer #3:

This article using 3 very interesting longitudinal data sets from Uganda to investigate how the acquisition of immunity is influenced by exposure and age. These are very pertinent questions. Unfortunately, the authors do not make good use of the excellent longitudinal data they have nor is the way they combine active and passive case detection samples in the definitions of anti-parasite and anti-disease very clear.

The authors treat all infections within a child as statistically independent. This is not strictly speaking correct.

Firstly, infections within the same child are controlled by the same immune system and thus reflective of the same immune status. Appropriate analyses of these data would thus need to take into account the non-independence of samples within the same individual.

Secondly, asymptomatic *P. falciparum* may last for up to 1 year or more. Therefore, it is likely that in particular in at low transmission household 2 or more consecutive infection may represent a single *P. falciparum* infection. In low transmission settings where few infections are acquired asymptomatic infections are likely to be longer lasting both the more limited number of treatments and the lack of superinfections that will result in higher parasitaemias and clinical symptoms in their early (exponential) growth phase. This factor could explain the overall higher proportion of asymptomatic infections and the lower parasitaemia at a given in age at low transmission.

The authors have defined anti-disease immunity as difference in the mean objective temperature at a given parasitaemia. This may again be potentially impacted by the quite different numbers of observed symptomatic vs. asymptomatic detection samples. From the Materials and methods section it is not clear how author dealt with children that reports having been febrile in the last 24hrs but did not have an objective fever. Were there many such children (and how to the relate to listed asymtomatic infections) and how were they treated in the analysis. Also did the proportion of such children differ with age and exposure in similar ways that the proportion of asymptomatic infections differs (i.e. highest a low transmission, lowest at moderate transmission).

[Editors’ note: what now follows is the decision letter after the authors submitted for further consideration.]

Thank you for submitting your article "Quantification of anti-parasite and anti-disease immunity to malaria as a function of age and exposure" for consideration by *eLife*. Your article has been reviewed by three peer reviewers, including Ben Cooper as the Reviewing Editor and Reviewer #1, and the evaluation has been overseen by Prabhat Jha as the Senior Editor. The following individual involved in review of your submission has agreed to reveal their identity: James Watson (Reviewer #2).

The reviewers have discussed the reviews with one another and the Reviewing Editor has drafted this decision to help you prepare a revised submission.

Summary:

The authors sought to describe how both age and transmission intensity affect the dynamics of the development of immunity against symptomatic *Plasmodium falciparum* infection. They used data from 773 children in three cohort studies in Uganda (2011-2014), in settings with a wide range of transmission intensity (measured by the annual entomological inoculation rate (aEIR) is estimated from monthly mosquito household catch surveys). They used age and number of infectious bites per year to characterise how immunity develops over time. Immunity was assessed in two ways: anti-parasite immunity (measured by parasite density) and anti-disease immunity (measured by the parasite density which causes fever). The data were analysed using generalised additive models, which allow for complex relationships between aEIR, age and immunity. The results quantified how both age and exposure had important influences on immunity and suggested a surprising (and hard-to-explain) non-linear relationship where children living in the lowest transmission settings appeared to develop immunity faster than those in higher transmission settings.

Essential revisions:

This is a uniquely rich data set, that has the potential to yield important insights.

However, all reviewers were highly skeptical about some of the findings (as outlined in the reviews) which challenge most current thinking about malaria and immunity. Some of these findings are so surprising that we think more effort to check their robustness is needed.

One of the biggest concerns was the grouping of active and passive case detection data without regard to the different data generating mechanisms. In particular there may be important biases operating (particularly in the passive data, which accounts for the majority of the data).

For these reasons we would like to see the following revisions:

1a) Repeat the analysis just with the active case detection data (this should provide the basis for the most reliable inference).

1b) Compare 1a with the same analysis on passive case detection data.

If 1a and 1b give essentially the same results this provides a useful confirmation – if the same results can been shown with data collected in two different ways that should lead to more confidence in the results. If results are different, then some thought about the reasons why are needed (though this in itself could be an interesting finding).

2) The current models use a mean aEIR, but EIR can vary substantially over time (and there is clear evidence that it does vary greatly in this data set]. This means that the model is implicitly assuming that a person's immunity at one time point depends on what happens (to the EIR) in the future. This doesn't make a lot of sense. It should not be difficult to account for this temporal variation in the EIR in the analysis (i.e. considering cumulative EIR), and we consider this an essential revision.

3) All non-linear models should be compared to their linear counterpart. In the current submission AP1 is the linear version of AP2, but there is no linear version of AP3 with the age/EIR interaction (similarly for the ADx models that consider temperature).

4) Additional checking of model adequacy is needed e.g. plots of scaled residuals and predictive checks (see, for example, https://people.maths.bris.ac.uk/~sw15190/mgcv/check-select.pdf).

5) The evidence for the surprising non-monotonic relationships for acquired immunity predicted by the model would be greatly strengthened if the authors could present a plot clearly showing this signal in the data (i.e. without a statistical model).

Reviewer #1:

I reviewed an earlier submission of this manuscript which was rejected largely because of the consensus (following consultation between reviewers) that the analysis was not making the best of use of the data. In particular, not accounting for individual-level malaria history from previous visits was felt to be a big limitation.

There have been a number of changes in the current submission which have undoubtedly improved the manuscript and addressed many substantive concerns. First, the provision of the code and data needed for the analysis is very welcome and helps clarify exactly what was done and makes it possible for others to verify the findings.

Second, the revision has clarified that dependencies were accounted for (with a random effects term), and explained why more explicitly accounting for individual histories was not done (because the data did not come from a birth cohort and only limited windows of time were observed for each individual). It has also clarified that individual exposure histories are being implicitly accounted for through adjustment for age, aEIR and through the random effects terms. In light of these clarifications, the modelling decisions seem more understandable.

In addition to these clarifications there is some new material including individual immunity trajectories (output from original model), additional models which did use data on recent exposure (models adjusted for number of *P. falciparum* positive visits in the previous 3 and 6 months periods), but found no evidence that this recent exposure was associated with either anti-parasite or anti-disease immunity (when already adjusting for age, aEIR and random effects).

There is also some additional sensitivity analysis e.g. adjusting for cumulative aEIR (rather than just aEIR) which suggests an effect of age independent of exposure.

Finally, the authors have added figures showing confidence bounds for age and EIR-specific anti-parasite and anti-disease immunity. This were omitted from the original submission and are also a welcome addition.

As the rebuttal letter makes clear, the authors do not have a clear explanation for the "provocative finding" that children exposed to the lowest transmission are developing immunity more effectively than those in more moderate transmission settings (although the parasite diversity hypothesis at least sounds plausible). Clearly, this finding goes against most preconceptions about how immunity in malaria works and I think the authors are therefore right to spend a lot of effort in sensitivity analyses to show that this is a robust finding and not an artefact. While the fact that same effect is seen for two different measures of immunity strengthens the case for this being a real effect, I am concerned that there are other possible reasons why this could still be an artefact which have not yet been ruled out. Two possibilities I can think of are:

i) Is there a possibility that differences in measuring aEIR at different sites could account for this? The sensitivity analysis excluding Walukuba helps here (Figure 5—figure supplement 5). What about looking at Kihihi/Kanungu alone? This seems to have a big enough range of aEIRs that the effect should be seen if present. [note: following consultation this was not considered to be an essential revision].

ii) Is there a possibility that there are edge effects in fitting that GAMs that introduce these effects? This could be checked, for example, by fitting the same models to simulated data under different assumptions. Is it possible get similar results (showing immunity developing faster at high and low aEIRs than at intermediate values) even when data are simulated under the assumption of a monotonic relationship between aEIR and the rate at which immunity develops? [note: following consultation this was not considered an essential revision].

When comparing the model fits (Supplementary file 1), the flexible GAM models (which allow for non-monotonic relationships between aEIR & immunity) are compared with a single mixed model expressing a monotonic relationship (with temperature and log parasite density varying linearly with age and with log(aEIR)). Why not also consider other functional monotonic relationships between these measures of immunity and age and aEIR? [note: following consultation this was not considered an essential revision]

Finally, the mgcv packages allows a lot of flexibility when fitting GAMS/GAMMs including allowing for autocorrelated errors. Without diving into the code (which I haven't done) it is not clear what choices were made and why form the supplementary material alone. Are key results robust to modelling decisions made, and what diagnostics checks have been done?

*Reviewer #2:*

This is a data analysis exercise using cohort studies taken across the spectrum of Pf transmission in Africa. They use age and number of infectious bites per year to characterise how immunity develops over time. This is a very nice dataset and overall the paper is well written, and the plots are informative and clear. Code and data were provided online in order to reproduce the results.

I have a few concerns in the following paragraphs.

“aEIR as a metric of individual exposure”:

The annual entomological inoculation rate (aEIR) is estimated from monthly mosquito household catch surveys. This estimated quantity is then plugged into the models (on the log scale) as the proxy for the total number of infectious bites per year. Given that in the lowest transmission setting, the median number of infectious bites is 2 per year, this implies that a median of 2 infectious mosquitoes were captured over the course of one year. What is the median denominator here, e.g. how many mosquitoes were captured? The uncertainty on these aEIR estimates must be much larger for the low transmission settings vs high transmission settings. This may explain why the R2 is also much lower when correlating log(aEIR) and hazard of infection (as calculated from a time-to-event model). This may also explain the trends observed in Figure 3 (top right panel), where the individuals with lowest aEIR have lower parasite densities that those with medium aEIR.

This potentially indicates that the aEIR is not a good estimate of the true number of infectious bites at very low transmission? If so, this means the conclusions drawn from the models cannot be taken at face value. [note: following consultation the consensus was that EIR is not thought to be problematic, though models in which future values of EIR influence the present are considered to be unjustifiable (since events in the future shouldn't influence the present)]

Model comparison:

The online code was broken in parts and I had to spend an hour debugging it. This is quite annoying!

My main question was to understand whether the more complex non-linear models (GAMs) were truly better than their linear counterparts. In particular whether the pattern shown in Figure 5B was not the result of over-fitting. If I understand Figure 5B correctly (probably the most important plot in the paper), a 2 year old with an aEIR of 2 infectious bites per year has on average the same anti-disease immunity as a 5 year old with an aEIR of ~200 infectious bites per year. This is very hard to believe and my prior on this being true would be very low. I believe this may be an artefact due to the problem highlighted above for the extremely low aEIRs.

I ran a series of 50 random 10-fold cross-validation experiments in order to characterize the out-of-sample mean squared error of each of the models 1-3 and a fourth model: the linear equivalent of model 3 (interaction between age and log(aEIR) with random effects at the individual and household level). The author's model 2 (GAM, no interaction term) performs best on average but the effect is very small. The fact that the non-linear model fits best the data is driving the surprising conclusions and I remain skeptical regarding the following line of the Abstract:

"Our findings suggest a strong effect of age on both types of immunity, that is not explained by cumulative exposure. They also show a non-linear effect of transmission intensity, where children living in the lowest transmission intensity setting appear to develop immunity faster than those experiencing higher transmission."

Even if the underlying relationship is truly non-linear, it would be useful to report the conclusions of a linear model which are likely to be generalisable. This would be similar to what the authors have done (subsection “Anti-parasite immunity”, fourth paragraph), but for instance reported as: for a twofold increase in yearly inoculation, the pyrogenic densities decrease by X amount.

Reviewer #3:

It's tough to come out against this one as the authors have used some great data sets and innovative statistical modelling approaches to tackle a very hard question, namely how is immunity to malaria acquired in moderate to high transmission settings. My concern is that the design of the data, and the design of the model don't fit well together leading to some hard to explain results.

Hard to explain results:

There are at least two results which I would put in the 'hard to explain' category.

Firstly, the authors have proposed biologically and epidemiologically well-informed hypotheses to explain the non-monotonic relationship with aEIR in Figure 7. However a simpler hypothesis is that this is an artefact of combining this particular model and this particular data set.

The second result is from Figure 6 where the fever threshold for 39C comes in and out of view as we increase from (a) aEIR = 2; (b) aEIR = 10; (c) aEIR = 50; to (d) aEIR = 200. This result would imply that a 2 year old with parasite density of 1e6 parasites/uL has fever > 39C when aEIR = 2; tolerates parasites better by having fever < 39C at aEIR = 10; then tolerates parasites worse by having fever > 39C when aEIR = 50; before finally switching back to better toleration of parasites with fever < 39C when aEIR = 200.

"They also show a non-linear effect of transmission intensity, where children living in the lowest transmission intensity setting appear to develop immunity faster than those experiencing higher transmission."

In an Abstract, one has little choice but to read a statement literally. This implies that children living in the lowest transmission setting appear to develop immunity faster than those experiencing higher transmission. This flies in the face of conventional malaria epidemiology. Perhaps the authors are trying to say something more subtle like there is a diminishing contribution to the acquisition of immunity with subsequent infections.

Model:

The authors have provided a change in notation to emphasise the utilisation of random effects, but otherwise left the model and results unchanged. The random effects at the individual and household level will account for a substantial degree of the variation between children. However, within an individual the only piece of longitudinal information included is age, incremented by the index k for visit. Otherwise, incidence of symptomatic episodes are independent (and there can be up to 30 of them). For example, the probability of symptomatic malaria is the same for a child on their 1st and 10th episodes.

It is notable that although age is updated at each visit, aEIR is not – instead being averaged over the duration of follow-up. Looking at Figure 5 of the original publication by Kamya et al., we see that there is a lot of time-dependent variation. For example, in Kihihi there is a huge spike in aEIR in 2013 towards the end of the study.

Some of the 'hard to explain' results could be arising from non-linearities and tensor interactions in the generalised additive models. Looking at models AP1-3, I note very little change in% deviance explained despite changes in the AIC. Looking at models SM1-3, I note that as model complexity increases, the% deviance explained actually decreases, despite the AIC appearing to favouring the more complex models.

Data:

I am concerned that the passively detected cases and routinely (every 3 months) detected cases are fundamentally incompatible.

If I interpret things correctly, a passively detected case will by definition contribute 1 count of a symptomatic episode and 1 light-microscopy measurement of parasite density. These data provide no discriminatory power on whether an infection (upon becoming microscopically detectable) becomes symptomatic, and no information on temperatures < 38°C.

By not having the precondition of treatment-seeking behaviour and fever >38°C, routinely detected cases arguably provide richer information. Although the majority of these measurements are parasite negative (by microscopy) and presumably excluded.

In Table 1 there are reported to be 2447 + 1555 + 207 = 4209 symptomatic episodes versus 955 + 331 + 145 = 1431 asymptomatic parasitaemia episodes. This suggests that the passively detected cases are provided far more data points.

[Editors' note: further revisions were requested prior to acceptance, as described below.]

Thank you for resubmitting your work entitled "Quantification of anti-parasite and anti-disease immunity to malaria as a function of age and exposure" for further consideration at *eLife*. Your revised article has been favorably evaluated by Prabhat Jha (Senior Editor), and three reviewers, one of whom is a member of our Board of Reviewing Editors.

The reviewers noted that the authors have worked hard to address the raised concerns, generating new sensitivity analyses that lend supporting evidence to their conclusions and providing new presentations of the data to better visualise the non-monotonic relationships.

The authors have now adequately addressed all but one of the reviewers' main concerns.

The one last point, which reviewer 3 noted the authors did not address in their previous response, was related to the unexpected result of% Deviance Explained as the complexity increases for the nested models for anti-parasite immunity and overall immunity against symptomatic malaria. From the previous review:

"Looking at models AP1-3, I note very little change in% deviance explained despite changes in the AIC. Looking at models SM1-3, I note that as model complexity increases, the% deviance explained actually decreases, despite the AIC appearing to favouring the more complex models."

This is an important point that requires a response.

Reviewer 2 also made the comment that It would be useful if the authors provided an extra variable in their data on github: passive versus active detection.

---

## [Author Response]

[Editors’ note: the author responses to the first round of peer review follow.]

We would like to submit a new version of our manuscript “Quantification of anti- parasite and anti-disease immunity to malaria as a function of age and exposure” for publication in *eLife*.

This manuscript was previously submitted to *eLife* and rejected due to specific concerns from the reviewers and editors. In particular, reviewers thought that our analysis failed to account properly for individual exposure history, and that it made certain unjustifiable assumptions. We have now revised our analysis to address these and other concerns expressed in the reviews and to clarify our assumptions and rationale for the analyses presented.

In summary, the major changes made were:

We have performed additional analyses where we use the rich data on recent exposure to *P. falciparum* to assess whether recent exposure is associated with different levels of anti-parasite and anti-disease immunity;

To show that our analysis captures individual heterogeneity in immunity, we have added a figure showing the predicted trajectories of anti-parasite and anti- disease immunity for a subset of individuals, as a function of age;

We have modified the Materials and methods section to make clear that our analyses do explicitly consider the longitudinal nature of the data. All our models include random effects at the individual and household levels to account for the lack of independence at both levels;

We have added extensive sensitivity analyses to show that our main findings are robust to the assumptions made;

We have made all the data used for these analyses, as well as the R code used to reproduce the main study findings available in our Github repository.

We want to underscore that we believe that the metric of individual exposure used in this analysis (annual entomological inoculation rate, aEIR) is a major, unique strength of this study. While no metric of natural exposure is perfect, our frequent, household level mosquito captures could be considered the most proximal feasible metric to the ideal when modeling the effect of exposure (transmission intensity) on malaria immunity. It directly quantifies the extent of environmental exposure to malaria, it is completely independent from the outcome of interest (immunity), and is strongly correlated with the individual hazard of infection.

As stated in our initial submission, we think that our paper should be considered for publication in *eLife* because it represents a major advance in our understanding of how immunity to malaria develops across a broad transmission spectrum. Improved understanding of the factors driving anti-malarial immunity is fundamental to investigate underlying biological mechanisms and to anticipate the impact of malaria control interventions. Careful analyses of longitudinal datasets, like this one, are required to fill existing gaps. Given its broad readership, we feel that *eLife* is the ideal forum for publication of these findings.

Reviewer #2:This study analyses data from three longitudinal cohort studies from regions of Uganda with varying transmission intensity. The analysis is based on very detailed epidemiological data sets with data collected on P. falciparum parasite density, temperature, clinical episodes of malaria and EIR. In order to assess the statistical methods it is worthwhile to consider this analysis in the context of a previous analysis of Ugandan data by the same authors in an excellent paper published last year in The Journal of Infectious Diseases. The data utilised in this analysis is superior to that utilised in the previous paper in a number of ways: more participants; household measurements of EIR; individual level measurements of temperature and parasite density, etc. The absence of data on low density PCR-detectable infections however is a limitation. Although the statistical models utilised in this analysis are justifiable and robust, I feel that the methods have taken a step back compared to their previous methods.In their previous analysis the authors captured how previous exposure to malaria affects the probability that an infection will lead to a clinical episode (e.g. individuals with a greater number of episodes in the past have greater levels of clinical immunity and thus a lower probability that a new infection progresses to an episode of clinical malaria). It appears to me that in this analysis that the rich data on an individuals' exposure history is not accounted for – and with a mean of 2 years' follow-up (over 3 years in some case) this is quite substantial data. Instead, the authors' models for temperature, parasite density and symptomatic malaria for visit k, depend only on measurements at visit k, and not on the individuals' malaria history at previous visits. I would like to see the authors utilise a method that accounts for malaria history at visits before k (e.g. such as the method used in their previous analysis).

We thank the reviewer for their thoughtful comments. There are significant differences between the previous (JID) and current datasets that guided the choice made around each analysis.

First, the dataset used in the previous analysis involved data from a birth cohort that followed participants between 6 months and 5 years of age. This cohort was characterised by extremely frequent follow-up, with blood smears performed at least every month on every trial participant (and every week during the first month after each malaria treatment). Thus, in this dataset we were confident that we had observed nearly all symptomatic malaria infections experienced by these children since birth (as well as a fraction of asymptomatic infections), which allowed us to use the observed prior malaria history as a metric of prior malaria exposure. In contrast, the PRISM cohorts used in this analysis are not birth cohorts (children aged 6 months to 10 years were enrolled) and regular visits occurred every 3 months. For each individual we only observe a limited window of time, (that varies depending on the age at enrollment) and we therefore lack the full malaria history for most individuals.

Second, a limitation of the approach used in the previous analysis is that as immunity develops, incidence of symptomatic malaria becomes a poor metric of past exposure, as a larger proportion of infections remain asymptomatic. Thus, the outcome of interest (immunity) directly affects the metric of exposure. This is particularly important in the PRISM study, as children were followed up to age 11 and had developed substantial immunity by then. An important strength of the PRISM study, used in the current analysis, is that it includes monthly mosquito collections, thus providing a metric of individual exposure (annual entomological inoculation rate) that is fully independent from the outcome of interest (immunity). The average entomological inoculation rate is the metric of exposure that we used for the analysis presented here.

Third, while the dataset from the previous analysis only included data on 100 children living in a single district, the PRISM cohort followed over 900 kids across settings with different transmission. The richness of the PRISM dataset allowed us to go beyond simply modeling the probability of symptomatic disease upon infection (as done in the JID paper) and model anti-parasite and anti- disease immunity independently. Furthermore, we were able to model non-linear relationships between the predictors and outcomes of interest. Thus, the present analysis is a significant advance over our previous analysis.

While it is true that in the current analysis we do not adjust directly for the cumulative number of episodes experienced by each individual (because as stated above, we don’t have such information), it does account for individual exposure histories in two ways:

The models are adjusted for age, aEIR, and their interaction (the product of age and EIR can be thought of as a metric of cumulative exposure), and therefore explicitly allow for greater immunity in children that have been exposed more;

The models contain random effects at the individual level, thus accounting for correlation between multiple observations of an individual.

We want to underscore that we believe that the annual entomological inoculation rate (aEIR), used as the metric of individual exposure in this analysis is actually a major, unique strength of this study. While no metric of natural exposure is perfect, our frequent, household level mosquito captures could be considered the most proximal feasible metric to the ideal when modeling the effect of exposure (transmission intensity) on malaria immunity. It directly quantifies the extent of environmental exposure to malaria, it is completely independent from the outcome of interest (immunity), and has been shown to be correlated with the individual hazard of infection (Figure 2).

Changes made:

To show that the model does capture individual trajectories in immunity (in a similar way as the JID paper did), we have added a paragraph to the Results section and a supplementary figure showing the predicted trajectories of anti-parasite and anti-disease immunity for a subset of individuals (Figure 5—figure supplement 12).

Furthermore, we have also performed an additional analysis where we use the rich data on recent exposure to malaria to assess whether recent exposure to malaria was associated with different levels of anti-parasite and anti-disease immunity, after adjusting for age and aEIR. These results are now presented in the Supplementary file 2 and Figure 5—figure supplement 3). We found no association between the number of recent *P. falciparum* infections and our outcomes of interest.

Finally, we have included an additional figure to the supplement showing the output of models adjusted for age and cumulative aEIR (instead of just aEIR) (Figure 5—figure supplement 2). As stated in the text, these results are consistent with a strong effect of age that is independent of exposure.

Reviewer #3:This article using 3 very interesting longitudinal data sets from Uganda to investigate how the acquisition of immunity is influenced by exposure and age. These are very pertinent questions. Unfortunately, the authors do not make good use of the excellent longitudinal data they have nor is the way they combine active and passive case detection samples in the definitions of anti-parasite and anti-disease very clear.

We regret that the reviewer feels this way about our analysis but feel that they have not fully understood our methods. We have made efforts (detailed further below) to add clarity and reduce the probability that readers will have similar concerns. The analyses all explicitly consider that the datasets are longitudinal and include random effects at the individual and household levels. Furthermore, we were explicitly concerned about how to properly combine active and passive case detection samples and performed sensitivity around this (adjusting for the probability of observation). Below, we provide answers to their specific concerns.

The authors treat all infections within a child as statistically independent. This is not strictly speaking correct.Firstly, infections within the same child are controlled by the same immune system and thus reflective of the same immune status. Appropriate analyses of these data would thus need to take into account the non-independence of samples within the same individual.

We fully agree with the reviewer that samples of the same individual can’t be treated as independent and apologize if this is not clear in the paper. All of our models include a random effect for each individual to account for the lack of independence. Furthermore they also include a random effect for each household to account for potential lack of independence at this level.

Changes made:

We have changed the text both in the Materials and methods section and in the supplementary material to make this more clear. The equations now include both individual and household random effects. Furthermore, the text now reads “To account for lack of independence, all models included random effects at the individual (u_i_) and household (γi) levels”.

Secondly, asymptomatic P. falciparum may last for up to 1 year or more. Therefore, it is likely that in particular in at low transmission household 2 or more consecutive infection may represent a single P. falciparum infection.

We agree with the reviewer that we can’t distinguish whether consecutive asymptomatic infection represent persistent infections, new infections or a combination. That is why we performed sensitivity analyses, where we accounted for the probability of observation (see Figure 5—figure supplement 6).

To further address this point, we have now performed an additional sensitivity analysis, were we only include “incident” asymptomatic infections, under the assumption that all subsequent asymptomatic samples represent a persistent (rather than a new infection) (Figure 5—figure supplement 7).

Changes made:

We have changed the text in the sensitivity analysis section as follows:

“Our main analyses include data from all visits regardless of their type (routine vs. passive case detection). […] Results from these analyses were qualitatively identical to the main analysis reported here and are presented in the supplementary material (Figure 5—figure supplements 6 and 7).”

We have included the results of the additional sensitivity analysis in the main text.

In low transmission settings where few infections are acquired asymptomatic infections are likely to be longer lasting both the more limited number of treatments and the lack of superinfections that will result in higher parasitaemias and clinical symptoms in their early (exponential) growth phase. This factor could explain the overall higher proportion of asymptomatic infections and the lower parasitaemia at a given in age at low transmission.

We agree with the reviewer that a lower rate of superinfections could explain why we observe higher “immunity” in lower transmission setting, particularly if coupled with lower parasite diversity. This is one of the hypothesis that we consider in our Discussion. To formally assess how the duration and complexity of infection affects clinical immunity, in our current studies we are genotyping all infections to follow them at the level of the individual strain.

Changes made:

For clarity, we have modified the paragraph discussing potential hypothesis surrounding this finding as follows:

“Lower complexity of infection coupled with lower parasite diversity in Walukuba, for example, could cause this difference, as developing an effective immune response against fewer parasite strains may be much easier than developing immunity against a more diverse pool (19,20). Testing this hypothesis would require careful characterization of the complexity and diversity of infections in each of our cohort settings.”

The authors have defined anti-disease immunity as difference in the mean objective temperature at a given parasitaemia. This may again be potentially impacted by the quite different numbers of observed symptomatic vs. asymptomatic detection samples.

The reviewer is right in stating that the different numbers of observed symptomatic/asymptomatic infections may have an impact on our results. To explore this potential source of bias, we performed sensitivity analyses where we weighted each asymptomatic infection based on the expected number of infections for a given period of time. These weights were calculated using a time-to-event model that estimated hazards of infection for each individual. Details of the method used and results are provided in the text.

From the Materials and methods section it is not clear how author dealt with children that reports having been febrile in the last 24hrs but did not have an objective fever. Were there many such children (and how to the relate to listed asymtomatic infections) and how were they treated in the analysis.

All of the children captured through the passive surveillance system who reported fever in the last 24h hours were tested for malaria, but only a fraction had objective fever when the temperature was measured.

The anti-disease model used objective temperature as the outcome, regardless of whether the child reported fever. While it is known that both parasite densities and temperature vary substantially over a 24h period, there is a very strong association between parasite density and temperature (as shown in Figure 6—figure supplement 1) and therefore we consider this to be a novel and appropriate outcome to assess anti-disease immunity.

We do acknowledge that by not taking into account the history of fever, whilst allowing a more specific and objective definition, we are losing some information on subjective symptoms. That’s why we also fit the “overall immunity model”, that does take into account all symptomatic malaria episodes, independent of the objective temperature recorded at the time of the visit (Figure 7).

Also did the proportion of such children differ with age and exposure in similar ways that the proportion of asymptomatic infections differs (i.e. highest a low transmission, lowest at moderate transmission).

We apologize but we do not fully understand the reviewer’s question. The prevalence of asymptomatic infections was highest in high transmission (nagongera) and lowest in moderate transmission as shown in Table 1 and Figure 1. Consistently, the proportion of smear positive children that had objective fever (at the time of the visit) was lowest in high transmission settings and highest in moderate transmission settings.

The prevalence of asymptomatic infection and the proportion of children with objective fever were higher and lower, respectively in children living in low transmission than in those in moderate transmission. These findings are consistent with higher levels of “anti-parasite” and “anti-disease” immunity as suggested by Figures 4-6 in the paper.

[Editors' note: the author responses to the re-review follow.]

Essential revisions:This is a uniquely rich data set, that has the potential to yield important insights.However, all reviewers were highly skeptical about some of the findings (as outlined in the reviews) which challenge most current thinking about malaria and immunity. Some of these findings are so surprising that we think more effort to check their robustness is needed.

We thank the reviewers and editors for their thorough and thoughtful review of our manuscript. When we planned this analysis, our main objective was to quantify the effects of age and exposure on the different types of immunity and we were also expecting to find a monotonic relationship between our exposures and outcomes of interest. We were equally (or more) surprised as the reviewers by the finding of the non-linearity uncovered by the analyses.

Even though we don’t have any conclusive evidence to determine what is driving this observed non-linearity, we are confident that it is not an artifact of our analysis. Children living in the lowest transmission setting (aEIR <5), and particularly children living in Walukuba, present with lower parasite densities and tolerate higher parasite densities than children living in higher transmission settings. This is evident both in the raw data (see response to Essential revision # 5) and in the results from the model.

While we are confident that this non-linearity exists in our data, as stated in the discussion we can’t rule out whether this finding reflects unmeasured differences between sites, as most of the observations of aEIR <5 come from a single site. Thus, it is possible, for example, that a lower parasite diversity in Walukuba could create this non-linearity. It is also possible that the children in the lowest transmission intensity are consistently experiencing less complex infections, that are ultimately easier to control than the more complex infections experienced in higher transmission settings. Additional potential explanations are presented in the Discussion.

We want to underscore that while this non-linearity is certainly counterintuitive and intriguing, the main purpose of our paper was to quantify the independent association between age, exposure and the different types of immunity. These rich datasets, from three cohorts conducted in parallel, provide a unique opportunity to address this question. In an effort to emphasize these results (and as suggested by reviewer #2), we have added a table (Table 2) summarizing the results of linear models. We report results from models fitted to the full dataset as well as models fit to observations from settings in which aEIR ≥5, where the relationship does seem to be linear, and for aEIR <5.

One of the biggest concerns was the grouping of active and passive case detection data without regard to the different data generating mechanisms. In particular there may be important biases operating (particularly in the passive data, which accounts for the majority of the data).For these reasons we would like to see the following revisions:1a) Repeat the analysis just with the active case detection data (this should provide the basis for the most reliable inference).

We disagree that the active case detection data provides the basis for the most reliable inference. Most malaria infections are symptomatic in this young population, and symptomatic infections are largely (>90%) captured by the passive surveillance system. The active surveillance system only captures symptomatic infections when the malaria episode happens to coincide with the already planned visit. Therefore, by excluding data from the passive surveillance system we would be excluding the vast majority of infections, and the most relevant outcomes for a study that aims to characterize immunity against symptomatic disease. Excluding the data from passive case detection would imply excluding >75% of observed infections in children <5 years and would bias our results towards lower density infections (i.e.; greater anti-parasite immunity).

It is true that an “ideal” study design would capture all infections soon after they occur and ascertain their outcome (symptomatic or not) by following them through time. However, this would probably require daily/weekly “active” sampling, particularly in the high transmission settings, which is not feasible. Our study design, by involving both passive and active case detection, aims to recreate this as closely as possible. Passive case detection is likely to capture the vast majority of infections that are symptomatic (>95% by self-report), while the active case detection captures whether an asymptomatic infection is present at the end of the period (this is more relevant as children develop immunity).

We agree that, given the differences in the two systems, we are more likely to observe a symptomatic than an asymptomatic infection and this could generate some bias in our results. That’s why we have performed sensitivity analyses (Figure 5—figure supplement 6) where we weight the observations from the active vs. passive systems to compensate for differences in the observation probabilities. We have also performed sensitivity analysis that is restricted to “incident” infections (Figure 5—figure supplement 7), to assess bias potentially introduced by persistent (rather than new) asymptomatic infections. None of these sensitivity analyses changed our inference substantially.

The reviewers suggest repeating the analyses with just the active and passive data. However, the fact that the two systems capture different types of infections (symptomatic vs. asymptomatics) also means that this is not possible for several of the analyses presented here. For example, we can’t fit the model on “overall immunity against symptomatic malaria” with just the passive data because by definition all of these infections are symptomatic. Similarly, we can’t estimate the fever threshold (anti-disease immunity) using just active data because there are essentially no objective fevers (temperature >38^o^C) captured by this system.

Where possible, we have fit the models with data from the active/passive system and results are shown below.

**Author response image 1. respfig1:** Anti-parasite immunity. Results of models quantifying anti-parasite immunity. Left panel shows result when models are fit to data from routine visits only (Active) while right panel shows result when models are fit to non-routine (Passive) visits.

**Author response image 2. respfig2:** Anti-disease immunity. Results of models quantifying disease immunity when fit to data from non-routine (Passive) visits only. It is not possible to fit this model using only data from routine visits as the majority (~95%) of participants are asymptomatic during these visits and therefore it is not possible to estimate a fever threshold.

As stated above, it is not possible to fit the overall immunity against malaria models using just the active or passive data, because by definition these visits capture symptomatic and asymptomatic infections respectively.

1b) Compare 1a with the same analysis on passive case detection data.If 1a and 1b give essentially the same results this provides a useful confirmation – if the same results can been shown with data collected in two different ways that should lead to more confidence in the results. If results are different, then some thought about the reasons why are needed (though this in itself could be an interesting finding).

It is only possible to repeat the analysis with just the active and just the passive data for the “anti-parasite” immunity models (Author response image 1). As expected, the “passive” results are consistent with the results of models including all data. In contrast, the “active” results show the strong association with age but not the association with aEIR. It is unclear whether these differences are a consequence of the reduced power of this restricted dataset or of the bias introduced by censoring most of the higher density infections that are detected by the passive detection system as discussed above.

We want to underscore that the active and passive surveillance systems do not represent two distinct data collection mechanisms. Passive and active case detection are complementary systems to ensure that we capture as many of the infections (both symptomatic and asymptomatic) as possible.

2) The current models use a mean aEIR, but EIR can vary substantially over time (and there is clear evidence that it does vary greatly in this data set]. This means that the model is implicitly assuming that a person's immunity at one time point depends on what happens (to the EIR) in the future. This doesn't make a lot of sense. It should not be difficult to account for this temporal variation in the EIR in the analysis (i.e. considering cumulative EIR), and we consider this an essential revision.

For our analysis, we wanted to use a metric of exposure that could capture the “average” exposure to *P. falciparum* that participants have experienced over their life-time (and not just during the study period). Since we have no information about exposure prior to study enrollment, we chose to use the mean aEIR over the study period as a proxy for mean life-time yearly exposure. Thus, our analysis implicitly assumes that aEIR has been stable in the three study settings, beyond seasonal fluctuations. While this assumption seems to be reasonable for Nagongera and Kihihi, it is more questionable for Walukuba, where transmission declined considerably after 2011 (this is stated in the Discussion). Nevertheless, under this stability assumption, including “future” observations is not problematic and in fact improves our exposure metric, by accounting for the seasonal variation of aEIR over three calendar years.

We fully agree that if we knew what the true cumulative exposure of each of the participants was at the time of enrollment, it would make sense to use the observed cumulative exposure in the analyses (rather than the average over the entire period) as suggested by the reviewers. However, in the absence of this data, using *just* the observed cumulative aEIR would imply making arguably stronger assumptions. For example, if we model the data from the beginning of the study (August-December 2011) only considering the aEIRs measured during this period, we would implicitly be assuming that these (more noisy) aEIRs represent the exposure experienced by participants prior to August 2011.

We want to emphasize that the main objective of our analysis was to characterize the independent effects of exposure and age on the development of anti-parasite and anti-disease immunity. In the absence of information about the exposure experienced by participants prior to study enrollment, we believe that using the average aEIR is the closest approximation to our metric of interest.

Changes made:

We have added a sentence to the Materials and methods section to explicitly state that our model (and our chosen metric of exposure) assumes that transmission has been stable at the three study sites. The text now reads:

“We included the EIR as an average (time-invariant) covariate, as we were interested in modeling the impact of the average exposure to malaria over time on the development of clinical immunity. Therefore, our model implicitly assumes that malaria transmission has been relatively stable at these three sites.”

As suggested by the reviewers, we have included an additional sensitivity analysis where, instead of using the average aEIR for the whole study period, we use the average up to the observation time-point (to avoid including future observations) (Figure 5—figure supplement 11).

3) All non-linear models should be compared to their linear counterpart. In the current submission AP1 is the linear version of AP2, but there is no linear version of AP3 with the age/EIR interaction (similarly for the ADx models that consider temperature).

We thank the reviewers for this suggestion. We have included a model with the linear interaction between age/EIR (now models AP3, AD3 and SM3).

4) Additional checking of model adequacy is needed e.g. plots of scaled residuals and predictive checks (see, for example, https://people.maths.bris.ac.uk/~sw15190/mgcv/check-select.pdf).

We thank the reviewers for this suggestion. We have included some supplementary figures checking model adequacy (Figures 5—figure supplements 2 and 3).

5) The evidence for the surprising non-monotonic relationships for acquired immunity predicted by the model would be greatly strengthened if the authors could present a plot clearly showing this signal in the data (i.e. without a statistical model).

We thank the reviewers for their observation. The non-monotonic relationship for acquired immunity captured by the model is certainly evident in the raw data, and this is one of the reasons why we are confident that this result is not an artifact of the analysis. Evidence for this signal is currently presented in one of the tables and two of the figures in the manuscript that summarize the raw data:

- Table 1 shows that the prevalence of asymptomatic parasitemia was greater among children living in the lowest transmission setting (Walukuba) than in children living in the medium transmission setting (Kihihi).

- Figure 1 is consistent with Table 1 and shows that the prevalence of asymptomatic parasitemia was greater in children 1-6 years old living in Walukuba (lowest transmission) than in children of the same age living in Kihihi.

- Figure 3 (bottom right panel) shows the non-monotonicity most clearly. This figure shows that for a given parasite density, children living settings with the lowest aEIRs tend to develop lower objective temperatures than children living in slightly higher transmission setting. This is consistent with the non-monotonicity observed for anti-disease immunity. A similar trend is observed for parasite densities (anti-parasite immunity, top right).

To further show that the signal is present in the raw data, we have produced two additional figures to complement those presented in the manuscript. Author response image 3 is an adaptation of Figure 3 (left upper and lower panels) from the manuscript and shows that both anti-parasite (left) and anti-disease (right) immunity seem to develop faster in the lowest transmission setting (Walukuba) than in Kihihi. Author response image 4 is a recreation of Figure 4 from the manuscript, but summarizing the raw data rather than the model output. In it, it is evident that participants from the lowest transmission setting (aEIR <5) tend to develop lower parasite densities (anti-parasite immunity) and lower objective temperatures (anti-disease immunity) than children living in settings with higher aEIR (aEIR 8-20). These results are highly consistent with estimates from the best fitting model.

**Author response image 3. respfig3:** Trends in parasite densities recorded during symptomatic (passive surveillance) infections and routine (active surveillance) visits as a function of age; and trends in the objective temperature recorded during visits in which participants were found to have a parasite density between 50,000 parasites/μL and 200,000 parasites/μL, as a function of age Each point represents a measurement obtained during a study visit. Dark squares represent the mean of the observations from each site.

**Author response image 4. respfig4:** Figure summarizing evidence for anti-parasite immunity (left) and anti-disease immunity (right) in the raw data. Each plot shows, for specific ages and aEIRs, the mean parasite density (/μL) (left), and the mean objective temperature given a parasite density between 10,000 to 80,000 parasites/μL (right). Means and 95% confidence intervals are shown. This plot corresponds to Figure 4 in the manuscript but was created using raw data (rather than model output).

Reviewer #1:I reviewed an earlier submission of this manuscript which was rejected largely because of the consensus (following consultation between reviewers) that the analysis was not making the best of use of the data. In particular, not accounting for individual-level malaria history from previous visits was felt to be a big limitation.There have been a number of changes in the current submission which have undoubtedly improved the manuscript and addressed many substantive concerns. First, the provision of the code and data needed for the analysis is very welcome and helps clarify exactly what was done and makes it possible for others to verify the findings.Second, the revision has clarified that dependencies were accounted for (with a random effects term), and explained why more explicitly accounting for individual histories was not done (because the data did not come from a birth cohort and only limited windows of time were observed for each individual). It has also clarified that individual exposure histories are being implicitly accounted for through adjustment for age, aEIR and through the random effects terms. In light of these clarifications, the modelling decisions seem more understandable.In addition to these clarifications there is some new material including individual immunity trajectories (output from original model), additional models which did use data on recent exposure (models adjusted for number of P. falciparum positive visits in the previous 3 and 6 months periods), but found no evidence that this recent exposure was associated with either anti-parasite or anti-disease immunity (when already adjusting for age, aEIR and random effects).There is also some additional sensitivity analysis e.g. adjusting for cumulative aEIR (rather than just aEIR) which suggests an effect of age independent of exposure.Finally, the authors have added figures showing confidence bounds for age and EIR-specific anti-parasite and anti-disease immunity. This were omitted from the original submission and are also a welcome addition.As the rebuttal letter makes clear, the authors do not have a clear explanation for the "provocative finding" that children exposed to the lowest transmission are developing immunity more effectively than those in more moderate transmission settings (although the parasite diversity hypothesis at least sounds plausible). Clearly, this finding goes against most preconceptions about how immunity in malaria works and I think the authors are therefore right to spend a lot of effort in sensitivity analyses to show that this is a robust finding and not an artefact. While the fact that same effect is seen for two different measures of immunity strengthens the case for this being a real effect, I am concerned that there are other possible reasons why this could still be an artefact which have not yet been ruled out. Two possibilities I can think of are:i) Is there a possibility that differences in measuring aEIR at different sites could account for this? The sensitivity analysis excluding Walukuba helps here (Figure 5—figure supplement 5). What about looking at Kihihi/Kanungu alone? This seems to have a big enough range of aEIRs that the effect should be seen if present. [note: following consultation this was not considered to be an essential revision].

We do not think that differences in measuring aEIR could account for the difference, as all study procedures were standardized at all sites. The reviewer is right in pointing out that the non-linearity should be evident if the model is fit to just the data from Kihihi and Walukuba. Please see Author response image 5.

**Author response image 5. respfig5:** Results of models quantifying anti-parasite and anti-disease immunity. This figure is similar to figure 5 in the manuscript, but with the analysis limited to 2238/5640 observations from Walukuba and Kihihi (thus the limited aEIR range).

ii) Is there a possibility that there are edge effects in fitting that GAMs that introduce these effects? This could be checked, for example, by fitting the same models to simulated data under different assumptions. Is it possible get similar results (showing immunity developing faster at high and low aEIRs than at intermediate values) even when data are simulated under the assumption of a monotonic relationship between aEIR and the rate at which immunity develops? [note: following consultation this was not considered an essential revision].

We thank the reviewer for their observation. As stated above (Essential revision #5) the fact that children in the lowest transmission setting are more immune is evident in the raw data and in all analyses. Therefore, we do not think it is an edge effect.

When comparing the model fits (Supplementary file 1), the flexible GAM models (which allow for non-monotonic relationships between aEIR & immunity) are compared with a single mixed model expressing a monotonic relationship (with temperature and log parasite density varying linearly with age and with log(aEIR)). Why not also consider other functional monotonic relationships between these measures of immunity and age and aEIR? [note: following consultation this was not considered an essential revision]

We thank the reviewer for their observation. We have now included a model that includes a linear interaction between age and log(aEIR) (Models AP3, AD3 and SM3). The models allowing for non-linear relationships consistently fit the data better.

Finally, the mgcv packages allows a lot of flexibility when fitting GAMS/GAMMs including allowing for autocorrelated errors. Without diving into the code (which I haven't done) it is not clear what choices were made and why form the supplementary material alone. Are key results robust to modelling decisions made, and what diagnostics checks have been done?

We thank the reviewer for their observation. As described above (Essential revision #5) we are confident that our finding is not an artifact of the analyses. The non-monotonicity in the association between aEIR and the outcomes of immunity is evident in the raw data and is also in the simplest of models (linear models stratified by aEIR).

We have added some diagnostic plots to the supplementary figures.

Reviewer #2:This is a data analysis exercise using cohort studies taken across the spectrum of Pf transmission in Africa. They use age and number of infectious bites per year to characterise how immunity develops over time. This is a very nice dataset and overall the paper is well written, and the plots are informative and clear. Code and data were provided online in order to reproduce the results.I have a few concerns in the following paragraphs.“aEIR as a metric of individual exposure”:The annual entomological inoculation rate (aEIR) is estimated from monthly mosquito household catch surveys. This estimated quantity is then plugged into the models (on the log scale) as the proxy for the total number of infectious bites per year. Given that in the lowest transmission setting, the median number of infectious bites is 2 per year, this implies that a median of 2 infectious mosquitoes were captured over the course of one year. What is the median denominator here, e.g. how many mosquitoes were captured? The uncertainty on these aEIR estimates must be much larger for the low transmission settings vs high transmission settings. This may explain why the R2 is also much lower when correlating log(aEIR) and hazard of infection (as calculated from a time-to-event model). This may also explain the trends observed in Figure 3 (top right panel), where the individuals with lowest aEIR have lower parasite densities that those with medium aEIR.This potentially indicates that the aEIR is not a good estimate of the true number of infectious bites at very low transmission? If so, this means the conclusions drawn from the models cannot be taken at face value.[note: following consultation the consensus was that EIR is not thought to be problematic, though models in which future values of EIR influence the present are considered to be unjustifiable (since events in the future shouldn't influence the present)]

See response above (Essential revision #2)

Model comparison:The online code was broken in parts and I had to spend an hour debugging it. This is quite annoying!My main question was to understand whether the more complex non-linear models (GAMs) were truly better than their linear counterparts. In particular whether the pattern shown in Figure 5B was not the result of over-fitting. If I understand Figure 5B correctly (probably the most important plot in the paper), a 2 year old with an aEIR of 2 infectious bites per year has on average the same anti-disease immunity as a 5 year old with an aEIR of ~200 infectious bites per year. This is very hard to believe and my prior on this being true would be very low. I believe this may be an artefact due to the problem highlighted above for the extremely low aEIRs.I ran a series of 50 random 10-fold cross-validation experiments in order to characterize the out-of-sample mean squared error of each of the models 1-3 and a fourth model: the linear equivalent of model 3 (interaction between age and log(aEIR) with random effects at the individual and household level). The author's model 2 (GAM, no interaction term) performs best on average but the effect is very small. The fact that the non-linear model fits best the data is driving the surprising conclusions and I remain skeptical regarding the following line of the Abstract:"Our findings suggest a strong effect of age on both types of immunity, that is not explained by cumulative exposure. They also show a non-linear effect of transmission intensity, where children living in the lowest transmission intensity setting appear to develop immunity faster than those experiencing higher transmission."

We are extremely thankful with the reviewer for taking the time to run our code and perform this cross-validation and apologize for the bugs in the code. As stated in our response to the Essential revisions (particularly #5) we are confident that the signal is present in the data. However, we agree that this finding is surprising and that’s why we have devoted a significant part of the discussion presenting different hypotheses that may explain this finding.

Changes made:

We have changed the wording of the Abstract as follows:

“Our findings also show an independent effect of exposure, where children living in moderate and high transmission settings tend to develop immunity faster as transmission increases. Surprisingly, children living in the lowest transmission intensity setting appear to develop immunity more efficiently than those living in the moderate transmission setting.”

Even if the underlying relationship is truly non-linear, it would be useful to report the conclusions of a linear model which are likely to be generalisable. This would be similar to what the authors have done (subsection “Anti-parasite immunity”, fourth paragraph), but for instance reported as: for a twofold increase in yearly inoculation, the pyrogenic densities decrease by X amount.

We thank the reviewer for their observation. We fully agree that results from the linear model are useful and more interpretable. We have now included a table in the main text (Table 2) summarizing the results of the linear models (anti-disease, anti-parasite and overall immunity). While a linear model makes the most sense for aEIR ≥5, we also report coefficients estimated when using the full dataset (and, for completeness, when aEIR <5).

Reviewer #3:It's tough to come out against this one as the authors have used some great data sets and innovative statistical modelling approaches to tackle a very hard question, namely how is immunity to malaria acquired in moderate to high transmission settings. My concern is that the design of the data, and the design of the model don't fit well together leading to some hard to explain results.Hard to explain results:There are at least two results which I would put in the 'hard to explain' category.Firstly, the authors have proposed biologically and epidemiologically well-informed hypotheses to explain the non-monotonic relationship with aEIR in Figure 7. However a simpler hypothesis is that this is an artefact of combining this particular model and this particular data set.

Please see response to Essential revision #5.

The second result is from Figure 6 where the fever threshold for 39C comes in and out of view as we increase from (a) aEIR = 2; (b) aEIR = 10; (c) aEIR = 50; to (d) aEIR = 200. This result would imply that a 2 year old with parasite density of 1e6 parasites/uL has fever > 39C when aEIR = 2; tolerates parasites better by having fever < 39C at aEIR = 10; then tolerates parasites worse by having fever > 39C when aEIR = 50; before finally switching back to better toleration of parasites with fever < 39C when aEIR = 200.

We thank the reviewer for their observation. The main result that we want to highlight on Figure 6 is how the fever threshold (temperature of 38^o^C) changes as a function of aEIR. Children living in a setting with aEIR=2 seem to tolerate higher parasite densities than children living in settings with aEIR of 10, 50 or 200 without developing objective temperature >38^o^C.

We really don’t have enough data to properly estimate the parasite densities associated with a temperature of 39^o^C or greater. As such, the “hard to explain result” that the reviewer highlights is probably driven by the large variance of the predictions, particularly around the edges (Figure 6—figure supplement 2A, B).

Changes made:

To avoid confusion we have removed the 39°C contour from the plot.

"They also show a non-linear effect of transmission intensity, where children living in the lowest transmission intensity setting appear to develop immunity faster than those experiencing higher transmission."In an Abstract, one has little choice but to read a statement literally. This implies that children living in the lowest transmission setting appear to develop immunity faster than those experiencing higher transmission. This flies in the face of conventional malaria epidemiology. Perhaps the authors are trying to say something more subtle like there is a diminishing contribution to the acquisition of immunity with subsequent infections.

We thank the reviewer for their comment. We have modified the wording of the Abstract as follows:

“Our findings also show an independent effect of exposure, where children living in moderate and high transmission settings tend to develop immunity faster as transmission increases. Surprisingly, children living in the lowest transmission intensity setting appear to develop immunity more efficiently than those living in the moderate transmission setting.”

Model:The authors have provided a change in notation to emphasise the utilisation of random effects, but otherwise left the model and results unchanged. The random effects at the individual and household level will account for a substantial degree of the variation between children. However, within an individual the only piece of longitudinal information included is age, incremented by the index k for visit. Otherwise, incidence of symptomatic episodes are independent (and there can be up to 30 of them). For example, the probability of symptomatic malaria is the same for a child on their 1st and 10th episodes.

While it is true that age is the main longitudinal variable, the model includes an interaction between our metric of exposure (aEIR) and age, which implicitly accounts for cumulative exposure (a function of time exposed (or age in endemic settings) and aEIR). Thus, the probability of symptomatic malaria would not be the same for a child on their 1^st^ and 10^th^ episodes unless they occurred on the same day.

Furthermore, we did consider models that explicitly account for cumulative exposure and for recent exposure. These results are presented in Figure 5—figure supplements 2 and 3.

It is notable that although age is updated at each visit, aEIR is not – instead being averaged over the duration of follow-up. Looking at Figure 5 of the original publication by Kamya et al., we see that there is a lot of time-dependent variation. For example, in Kihihi there is a huge spike in aEIR in 2013 towards the end of the study.

Please see response to Essential revision #2.

Some of the 'hard to explain' results could be arising from non-linearities and tensor interactions in the generalised additive models. Looking at models AP1-3, I note very little change in% deviance explained despite changes in the AIC. Looking at models SM1-3, I note that as model complexity increases, the% deviance explained actually decreases, despite the AIC appearing to favouring the more complex models.Data:I am concerned that the passively detected cases and routinely (every 3 months) detected cases are fundamentally incompatible.If I interpret things correctly, a passively detected case will by definition contribute 1 count of a symptomatic episode and 1 light-microscopy measurement of parasite density. These data provide no discriminatory power on whether an infection (upon becoming microscopically detectable) becomes symptomatic, and no information on temperatures < 38°C.

Please see response to Essential revision #1. The reviewer is right in saying that a passively detected case will, by definition, contribute 1 count of a symptomatic episode and 1 measurement of parasite density. However, it is not true that the data provide no information on temperatures <38°C. 1946/3788 passive visits had objective temperatures <38°C.

By not having the precondition of treatment-seeking behaviour and fever >38°C, routinely detected cases arguably provide richer information. Although the majority of these measurements are parasite negative (by microscopy) and presumably excluded.

Please see response to Essential revision #1. While it is true that routine visits do not have the pre-condition of treatment-seeking behavior, fever >38^o^C is not a pre-condition of non-routine visits (at the time of the visit, when the sample is taken).

It is not right to think either that routine visits are unbiased or provide richer information. In this pediatric population, most of the malaria infections are symptomatic (particularly in children < 5 years), and these are largely captured by the passive system (>95% of febrile illnesses are captured by the passive system). Routine visits only capture symptomatic malaria when (by coincidence) the person becomes sick the day of the visit. Thus, by including only routine visits we would be excluding the majority of the data and of the outcomes of interest.

In Table 1 there are reported to be 2447 + 1555 + 207 = 4209 symptomatic episodes versus 955 + 331 + 145 = 1431 asymptomatic parasitaemia episodes. This suggests that the passively detected cases are provided far more data points.

The reviewer is right. As stated above and in response to Essential revision #1, the passive surveillance system captures most malaria infections in this populations, as the majority of infections are symptomatic.

[Editors' note: further revisions were requested prior to acceptance, as described below.]

The authors have now adequately addressed all but one of the reviewers' main concerns.The one last point, which reviewer 3 noted the authors did not address in their previous response, was related to the unexpected result of% Deviance Explained as the complexity increases for the nested models for anti-parasite immunity and overall immunity against symptomatic malaria. From the previous review:"Looking at models AP1-3, I note very little change in% deviance explained despite changes in the AIC. Looking at models SM1-3, I note that as model complexity increases, the% deviance explained actually decreases, despite the AIC appearing to favouring the more complex models."This is an important point that requires a response.

We thank the reviewer for their comment and apologize for not addressing this point before.

While it is expected that AIC should correlate well with the% deviance explained, the correlation is not perfect particularly when the change in model fit is small. As the reviewer points out, when comparing models SM2 and SM4, the% deviance explained decreases while the AIC changes very little (from 5373 to 5369). This suggests that SM4 (the more complex model) is not really an improvement over SM2 (the model without the interaction). When comparing models AP1-AP4 the% deviance explained does not change much, while the changes in AIC are larger (from 14052 to 14032) and we interpret this as evidence of a better fit of the more complex model. Larger changes in both AIC and% deviance explained, as those observed comparing models AD1-AD4, do provide stronger support in favor of the more complex model for this particular outcome.

Since the non-linear interaction models seem to fit the data better for both anti-disease and anti-parasite immunity models, we decided to present results from this model for all outcomes of interest in the paper. However, the main study findings are robust to model choice. The main results are presented in (Figures 5A and 7) for models AP2 and SM2.

Changes made:

Since our model choice relied on AIC more than on percent deviance explained, we have changed the text in the Materials and methods as follows:

“Best fitting models were selected based on Akaike’s Information Criterion, but changes in the percent deviance explained are also presented.”

In addition, we have added some text to the section “Overall immunity against symptomatic malaria” stating that model fit was similar for models SM2 and SM4.

“Models allowing smooth relationships, with or without for two-way interactions, fit the data equally well.”

Reviewer 2 also made the comment that It would be useful if the authors provided an extra variable in their data on github: passive versus active detection.

We apologize if this is not clear in the data. All visits where the “visittype” variable is “Not routine” are from passive case detection. All other values (“Enrollment”, “Routine” and “T-cell visit”) correspond to active detection.

We have added this information to the data dictionary.